# REPRESENTATIONAL DISSIMILARITY METRIC SPACES FOR STOCHASTIC NEURAL NETWORKS

**Lyndon R. Duong,**[*] **Jingyang Zhou,**[*] **Josue Nassar, Jules Berman, Jeroen Olieslagers, Alex H. Williams**
Center for Computational Neuroscience, Flatiron Institute, New York, NY, 10010
Center for Neural Science, New York University, New York, NY, 10003
{lyndon.duong, jingyang.zhou, alex.h.williams}@nyu.edu

## ABSTRACT

Quantifying similarity between neural representations—e.g. hidden layer activation vectors—is a perennial problem in deep learning and neuroscience research. Existing methods compare deterministic responses (e.g. artificial networks that lack stochastic layers) or averaged responses (e.g., trial-averaged firing rates in biological data). However, these measures of *deterministic* representational similarity ignore the scale and geometric structure of noise, both of which play important roles in neural computation. To rectify this, we generalize previously proposed shape metrics (Williams et al., 2021) to quantify differences in *stochastic* representations. These new distances satisfy the triangle inequality, and thus can be used as a rigorous basis for many supervised and unsupervised analyses. Leveraging this novel framework, we find that the stochastic geometries of neurobiological representations of oriented visual gratings and naturalistic scenes respectively resemble untrained and trained deep network representations. Further, we are able to more accurately predict certain network attributes (e.g. training hyperparameters) from its position in stochastic (versus deterministic) shape space.

## 1 INTRODUCTION

Comparing high-dimensional neural responses—neurobiological firing rates or hidden layer activations in artificial networks—is a fundamental problem in neuroscience and machine learning (Dwivedi & Roig, 2019; Chung & Abbott, 2021). There are now many methods for quantifying representational dissimilarity including canonical correlations analysis (CCA; Raghu et al., 2017), centered kernel alignment (CKA; Kornblith et al., 2019), representational similarity analysis (RSA; Kriegeskorte et al., 2008a), shape metrics (Williams et al., 2021), and Riemannian distance (Shahbazi et al., 2021) . Intuitively, these measures quantify similarity in the geometry of neural responses while removing expected forms of invariance, such as permutations over arbitrary neuron labels.

However, these methods only compare *deterministic representations*—i.e. networks that can be represented as a function $f : \mathcal{Z} \mapsto \mathbb{R}^n$, where $n$ denotes the number of neurons and $\mathcal{Z}$ denotes the space of network inputs. For example, each $z \in \mathcal{Z}$ could correspond to an image, and $f(z)$ is the response evoked by this image across a population of $n$ neurons (Fig. 1A). Biological networks are essentially never deterministic in this fashion. In fact, the variance of a stimulus-evoked neural response is often larger than its mean (Goris et al., 2014). Stochastic responses also arise in the deep learning literature in many contexts, such as in deep generative modeling (Kingma & Welling, 2019), Bayesian neural networks (Wilson, 2020), or to provide regularization (Srivastava et al., 2014).

Stochastic networks may be conceptualized as functions mapping each $z \in \mathcal{Z}$ to a probability distribution, $F(\cdot \mid z)$, over $\mathbb{R}^n$ (Fig. 1B, Kriegeskorte & Wei 2021). Although it is easier to study the representational geometry of the average response, it is well understood that this provides an incomplete and potentially misleading picture (Kriegeskorte & Douglas, 2019). For instance, the ability to discriminate between two inputs $z, z' \in \mathcal{Z}$ depends on the overlap of $F(z)$ and $F(z')$, and not simply the separation of their means (Fig. 1C-D). A rich literature in neuroscience has built on top of this insight (Shadlen et al., 1996; Abbott & Dayan, 1999; Rumyantsev et al., 2020). However,

---

[*]Equal contribution.

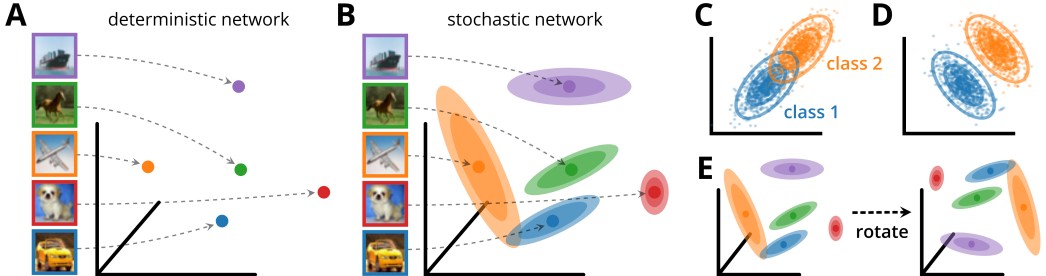

**Figure 1:** *(A)* Illustration of a deterministic network mapping inputs, (color-coded images) into points in $\mathbb{R}^n$. *(B)* Illustration of a stochastic network, where each input, $\boldsymbol{z} \in \mathcal{Z}$, maps onto a distribution, $F(\cdot \mid \boldsymbol{z})$, over $\mathbb{R}^n$. *(C)* Example where noise correlations impair discriminability between two image classes. *(D)* Example where noise correlations improve discriminability (see Abbott & Dayan, 1999). *(E)* Illustration of two stochastic networks with equivalent representational geometry.

to our knowledge, no studies have compared noise correlation structure across animal subjects or species, as has been done with trial-averaged responses. In machine learning, many studies have characterized the effects of noise on model predictions (Sietsma & Dow, 1991; An, 1996), but only a handful have begun to characterize the geometry of stochastic hidden layers (Dapello et al., 2021).

To address these gaps, we formulate a novel class of *metric spaces* over stochastic neural representations. That is, given two stochastic networks $F_i$ and $F_j$, we construct distance functions $d(F_i, F_j)$ that are symmetric, satisfy the triangle inequality, and are equal to zero if and only if $F_i$ and $F_j$ are equivalent according to a pre-defined criterion. In the deterministic limit—i.e., as $F_i$ and $F_j$ map onto Dirac delta functions—our approach converges to well-studied metrics over *shape spaces* (Dryden & Mardia, 1993; Srivastava & Klassen, 2016), which were proposed by Williams et al. (2021) to measure distances between deterministic networks. The triangle inequality is required to derive theoretical guarantees for many downstream analyses (e.g. nonparametric regression, Cover & Hart 1967, and clustering, Dasgupta & Long 2005). Thus, we lay an important foundation for analyzing stochastic representations, akin to results shown by Williams et al. (2021) in the deterministic case.

## 2 METHODS

### 2.1 DETERMINISTIC SHAPE METRICS

We begin by reviewing how shape metrics quantify representational dissimilarity in the deterministic case. In the *Discussion* (sec. 4), we review other related prior work.

Let $\{f_1, \ldots, f_K\}$ denote $K$ deterministic neural networks, each given by a function $f_k : \mathcal{Z} \mapsto \mathbb{R}^{n_k}$. Representational similarity between networks is typically defined with respect to a set of $M$ inputs, $\{\boldsymbol{z}_1, \ldots, \boldsymbol{z}_M\} \in \mathcal{Z}^M$. We can collect the representations of each network into a matrix:

$$\boldsymbol{X}_k = \begin{bmatrix} \rule{1em}{0.4pt} & f_k(\boldsymbol{z}_1) & \rule{1em}{0.4pt} \\ & \vdots & \\ \rule{1em}{0.4pt} & f_k(\boldsymbol{z}_M) & \rule{1em}{0.4pt} \end{bmatrix}. \tag{1}$$

A naïve dissimilarity measure would be the Euclidean distance, $\|\boldsymbol{X}_i - \boldsymbol{X}_j\|_F$. This is nearly always useless. Since neurons are typically labelled in arbitrary order, our notion of distance should—at the very least—be invariant to permutations. Intuitively, we desire a notion of distance such that $d(\boldsymbol{X}_i, \boldsymbol{X}_j) = d(\boldsymbol{X}_i, \boldsymbol{X}_j \boldsymbol{\Pi})$ for any permutation matrix, $\boldsymbol{\Pi} \in \mathbb{R}^{n \times n}$. Linear CKA and RSA achieve this by computing the dissimilarity between $\boldsymbol{X}_i \boldsymbol{X}_i^\top$ and $\boldsymbol{X}_j \boldsymbol{X}_j^\top$ instead of the raw representations.

*Generalized shape metrics* are an alternative approach to quantifying representational dissimilarity. The idea is to compute the distance after minimizing over nuisance transformations (e.g. permutations or rotations in $\mathbb{R}^n$). Let $\phi_k : \mathbb{R}^{M \times n_k} \mapsto \mathbb{R}^{M \times n}$ be a fixed, "preprocessing function" for each network and let $\mathcal{G}$ be a set of nuisance transformations on $\mathbb{R}^n$. Williams et al. (2021) showed that:

$$d(\boldsymbol{X}_i, \boldsymbol{X}_j) = \min_{\boldsymbol{T} \in \mathcal{G}} \|\phi_i(\boldsymbol{X}_i) - \phi_j(\boldsymbol{X}_j)\boldsymbol{T}\|_F \tag{2}$$

is a *metric* over equivalent neural representations provided two technical conditions are met. The first is that $\mathcal{G}$ is a *group* of linear transformations. This means that: (a) the identity is in the set of nuisance transformations ($\boldsymbol{I} \in \mathcal{G}$), (b) every nuisance transformation is invertible by another nuisance transformation (if $\boldsymbol{T} \in \mathcal{G}$ then $\boldsymbol{T}^{-1} \in \mathcal{G}$), and (c) nuisance transformations are closed under composition ($\boldsymbol{T}_1\boldsymbol{T}_2 \in \mathcal{G}$ if $\boldsymbol{T}_1 \in \mathcal{G}$ and $\boldsymbol{T}_2 \in \mathcal{G}$). The second condition is that every nuisance transformation is an *isometry*, meaning that $\|\boldsymbol{X}_i\boldsymbol{T} - \boldsymbol{X}_j\boldsymbol{T}\|_F = \|\boldsymbol{X}_i - \boldsymbol{X}_j\|_F$ for every $\boldsymbol{T} \in \mathcal{G}$. Several choices of $\mathcal{G}$ satisfy these conditions including the permutation group, $\mathcal{P}$, and the orthogonal group, $\mathcal{O}$, which respectively correspond to the set of all permutations and rotations on $\mathbb{R}^n$.

Equation (2) provides a recipe to construct many notions of distance. To enumerate some examples, we will assume for simplicity that $\phi_1 = \ldots = \phi_K = \phi$ and all networks have $n$ neurons. Then, to obtain a metric that is invariant to translations and permutations, we can set $\phi(\boldsymbol{X}) = (1/n)(\boldsymbol{I} - \boldsymbol{1}\boldsymbol{1}^\top)\boldsymbol{X}$ and $\mathcal{G} = \mathcal{P}(n)$. If we instead set $\mathcal{G} = \mathcal{O}(n)$, we recover the well-known *Procrustes distance*, which is invariant to rotations. Finally, if we choose $\phi(\cdot)$ to whiten the covariance of $\boldsymbol{X}$, we obtain notions of distance that are invariant to linear transformations and are closely related to CCA. Williams et al. (2021) provides further discussion and examples.

An attractive property of equation (2) is that it establishes a metric space over deterministic representations. That is, distances are symmetric $d(\boldsymbol{X}_i, \boldsymbol{X}_j) = d(\boldsymbol{X}_j, \boldsymbol{X}_i)$ and satisfy the triangle inequality $d(\boldsymbol{X}_i, \boldsymbol{X}_k) \leq d(\boldsymbol{X}_i, \boldsymbol{X}_j) + d(\boldsymbol{X}_j, \boldsymbol{X}_k)$. Further, the distance is zero if and only if there exists a $\boldsymbol{T} \in \mathcal{G}$ such that $\phi_i(\boldsymbol{X}_i) = \phi_j(\boldsymbol{X}_j)\boldsymbol{T}$. These fundamental properties are needed to rigorously establish many statistical analyses (Cover & Hart, 1967; Dasgupta & Long, 2005).

## 2.2 STOCHASTIC SHAPE METRICS

Let $\{F_1, \ldots, F_K\}$ denote a collection of $K$ stochastic networks. That is, each $F_k$ is a function that maps each input $\boldsymbol{z} \in \mathcal{Z}$ to a conditional probability distribution $F_k(\cdot \mid \boldsymbol{z})$. How can equation (2) be generalized to measure representational distances in this case? In particular, the minimization in equation (2) is over a Euclidean "ground metric," and we would like to choose a compatible metric over probability distributions. Concretely, let $\mathcal{D}(P, Q)$ be a chosen "ground metric" between two distributions $P$ and $Q$. Let $\delta_{\boldsymbol{x}}$ and $\delta_{\boldsymbol{y}}$ denote Dirac masses at $\boldsymbol{x}, \boldsymbol{y} \in \mathbb{R}^n$ and consider the limit that $P \to \delta_{\boldsymbol{x}}$ and $Q \to \delta_{\boldsymbol{y}}$. In this limit, we seek a ground metric for which $\mathcal{D}(\delta_{\boldsymbol{x}}, \delta_{\boldsymbol{y}})$ is related to $\|\boldsymbol{x} - \boldsymbol{y}\|$. Many probability metrics and divergences fail to meet this criterion. For example, if $\boldsymbol{x} \neq \boldsymbol{y}$, then the Kullback-Leibler (KL) divergence approaches infinity and the total variation distance and Hellinger distance approach a constant that does not depend on $\|\boldsymbol{x} - \boldsymbol{y}\|$.

In this work, we explored two ground metrics. First, the *p-Wasserstein distance* (Villani, 2009):

$$\mathcal{W}_p(P, Q) = (\inf \mathbb{E}\left[\|X - Y\|^p\right])^{1/p} \tag{3}$$

where $p \geq 1$, and the infimum is taken over all random variables $(X, Y)$ whose marginal distributions coincide with $P$ and $Q$. Second, the *energy distance* (Székely & Rizzo, 2013):

$$\mathcal{E}_q(P, Q) = (\mathbb{E}\left[\|X - Y\|^q\right] - \tfrac{1}{2}\mathbb{E}\left[\|X - X'\|^q\right] - \tfrac{1}{2}\mathbb{E}\left[\|Y - Y'\|^q\right])^{1/2} \tag{4}$$

where $0 < q < 2$ and $X, X' \overset{\text{i.i.d.}}{\sim} P$ and $Y, Y' \overset{\text{i.i.d.}}{\sim} Q$. As desired, we have $\mathcal{W}_p(\delta_{\boldsymbol{x}}, \delta_{\boldsymbol{y}}) = \|\boldsymbol{x} - \boldsymbol{y}\|$ for any $p$, and $\mathcal{E}_q(\delta_{\boldsymbol{x}}, \delta_{\boldsymbol{y}}) = \|\boldsymbol{x} - \boldsymbol{y}\|^{q/2}$. Thus, when $q = 1$ for example, the energy distance converges to the square root of Euclidean distance in the deterministic limit. Interestingly, when $q = 2$, the energy distance produces a deterministic metric on trial-averaged responses (see appendix F.1).

The Wasserstein and energy distances are intuitive generalizations of Euclidean distance. Both can be understood as being proportional to the amount of energy it costs to transport a pile of dirt (a probability density $P$) to a different configuration (the other density $Q$). Wasserstein distance is based on the cost of the optimal transport plan, while energy distance is based on the the cost of a random (i.e. maximum entropy) transport plan (see Supp. Fig. 1, and Feydy et al. 2019).

Our main proposition shows that these two ground metrics can be used to generalize equation (2).

**Proposition 1** (Stochastic Shape Metrics)**.** *Let $Q$ be a distribution on the input space. Let $\phi_1, \ldots, \phi_K$ be measurable functions mapping onto $\mathbb{R}^n$ and let $F_i^\phi = F_i \circ \phi_i^{-1}$. Let $\mathcal{D}^2$ denote the squared "ground metric," and let $\mathcal{G}$ be a group of isometries with respect to $\mathcal{D}$. Then,*

$$d(F_i, F_j) = \min_{\boldsymbol{T} \in \mathcal{G}} \left( \underset{\boldsymbol{z} \sim Q}{\mathbb{E}} \left[ \mathcal{D}^2\left( F_i^\phi(\cdot \mid \boldsymbol{z}), F_j^\phi(\cdot \mid \boldsymbol{z}) \circ \boldsymbol{T}^{-1} \right) \right] \right)^{1/2} \tag{5}$$

*defines a metric over equivalence classes, where $F_i$ is equivalent to $F_j$ if and only if there is a $\boldsymbol{T} \in \mathcal{G}$ such that $F_i^\phi(\cdot \mid \boldsymbol{z})$ and $F_j^\phi(\cdot \mid \boldsymbol{z}) \circ \boldsymbol{T}^{-1}$ are equal for all $\boldsymbol{z} \in supp(Q)$.*

Above, we use the notation $P \circ \phi^{-1}$ to denote the pushforward measure—i.e. the measure defined by the function composition, $P(\phi^{-1}(A))$ for a measurable set $A$, where $P$ is a distribution and $\phi$ is a measurable function. A proof is provided in appendix C. Intuitively, $\boldsymbol{T}$ plays the same role as in equation (2), which is to remove nuisance transformations (e.g. rotations or permutations; see Fig. 1E). The functions $\phi_1, \ldots, \phi_K$ also play the same role as "preprocessing functions," implementing steps such as whitening, normalizing by isotropic scaling, or projecting data onto a principal subspace. For example, to obtain a translation-invariant distance, we can subtract the grand mean response from each conditional distribution. That is, $\phi_k(\boldsymbol{x}) = \boldsymbol{x} - \mathbb{E}_{\boldsymbol{z} \sim Q}[\mathbb{E}_{\boldsymbol{x} \sim F_k(\boldsymbol{x} \mid \boldsymbol{z})}[\boldsymbol{x}]]$.

## 2.3 PRACTICAL ESTIMATION OF STOCHASTIC SHAPE METRICS

Stochastic shape distances (eq. 5) are generally more difficult to estimate than deterministic distances (eq. 2). In the deterministic case, the minimization over $\boldsymbol{T} \in \mathcal{G}$ is often a well-studied problem, such as linear assignment (Burkard et al., 2012) or the orthogonal Procrustes problem (Gower & Dijksterhuis, 2004). In the stochastic case, the conditional distributions $F_k(\cdot \mid \boldsymbol{z})$ often do not even have a parametric form, and can only be accessed by drawing samples—e.g. by repeated forward passes in an artificial network. Moreover, Wasserstein distances suffer a well-known curse of dimensionality: in $n$-dimensional spaces, the plug-in estimator converges at a very slow rate proportional to $s^{-1/n}$, where $s$ is the number of samples (Niles-Weed & Rigollet, 2022).

Thus, to estimate shape distances with Wasserstein ground metrics, we assume that, $F_i^\phi(\cdot \mid \boldsymbol{z})$, is well-approximated by a Gaussian for each $\boldsymbol{z} \in \mathcal{Z}$. The 2-Wasserstein distance has a closed form expression in this case (Remark 2.31 in Peyré & Cuturi 2019 and Theorem 1 in Bhatia et al. 2019):

$$\mathcal{W}_2(\mathcal{N}(\boldsymbol{\mu}_i, \boldsymbol{\Sigma}_i), \mathcal{N}(\boldsymbol{\mu}_j, \boldsymbol{\Sigma}_j)) = \left( \|\boldsymbol{\mu}_i - \boldsymbol{\mu}_j\|^2 + \min_{\boldsymbol{U} \in \mathcal{O}(n)} \|\boldsymbol{\Sigma}_i^{1/2} - \boldsymbol{\Sigma}_j^{1/2} \boldsymbol{U}\|_F^2 \right)^{1/2} \qquad (6)$$

where $\mathcal{N}(\boldsymbol{\mu}, \boldsymbol{\Sigma})$ denotes a Gaussian density. It is important not to confuse the minimization over $\boldsymbol{U} \in \mathcal{O}(n)$ in this equation with the minimization over nuisance transformations, $\boldsymbol{T} \in \mathcal{G}$, in the shape metric (eq. 5). These two minimizations arise for entirely different reasons, and the Wasserstein distance is not invariant to rotations. Intuitively, we can estimate the Wasserstein-based shape metric by minimizing over $\boldsymbol{U} \in \mathcal{O}(n)$ and $\boldsymbol{T} \in \mathcal{G}$ in alternation (for full details, see appendix D.1).

Approximating $F_i^\phi(\cdot \mid \boldsymbol{z})$ as Gaussian is common in neuroscience and deep learning (Kriegeskorte & Wei, 2021; Wu et al., 2019). In biological data, we often only have enough trials to estimate the first two moments of a neural response, and one may loosely appeal to the principle of maximum entropy to justify this approximation (Uffink, 1995). In certain artificial networks the Gaussian assumption is satisfied exactly, such as in variational autoencoders (see sec. 3.3). Finally, even if the Gaussian assumption is violated, equation (6) can still be a reasonable ground metric that is only sensitive to the first two moments (mean and covariance) of neural responses (see appendix E.3).

The Gaussian assumption is also unnecessary if we use the energy distance (eq. 4) as the ground metric instead of Wasserstein distance. Plug-in estimates of this distance converge at a much faster rate in high-dimensional spaces (Gretton et al., 2012; Sejdinovic et al., 2013). In this case, we propose a two-stage estimation procedure using iteratively reweighted least squares (Kuhn, 1973), followed by a "metric repair" step (Brickell et al., 2008) which resolves small triangle inequality violations due to distance estimation error (see appendix D.2 for full details).

We discuss computational complexity in Appendix D.1.1 and provide user-friendly implementations of stochastic shape metrics at: `github.com/ahwillia/netrep`.

## 2.4 INTERPOLATING BETWEEN MEAN-SENSITIVE AND COVARIANCE-SENSITIVE METRICS

An appealing feature of the 2-Wasserstein distance for Gaussian measures (eq. 6) is its decomposition into two terms that respectively depend on the mean and covariance. We reasoned that it would be useful to isolate the relative contributions of these two terms. Thus, we considered the following generalization of the 2-Wasserstein distance parameterized by a scalar, $0 \le \alpha \le 2$:

$$\overline{\mathcal{W}}_2^\alpha(P_i, P_j) = \left( \alpha \|\boldsymbol{\mu}_i - \boldsymbol{\mu}_j\|^2 + (2 - \alpha) \min_{\boldsymbol{U} \in \mathcal{O}(n)} \|\boldsymbol{\Sigma}_i^{1/2} - \boldsymbol{\Sigma}_j^{1/2} \boldsymbol{U}\|_F^2 \right)^{1/2} \qquad (7)$$

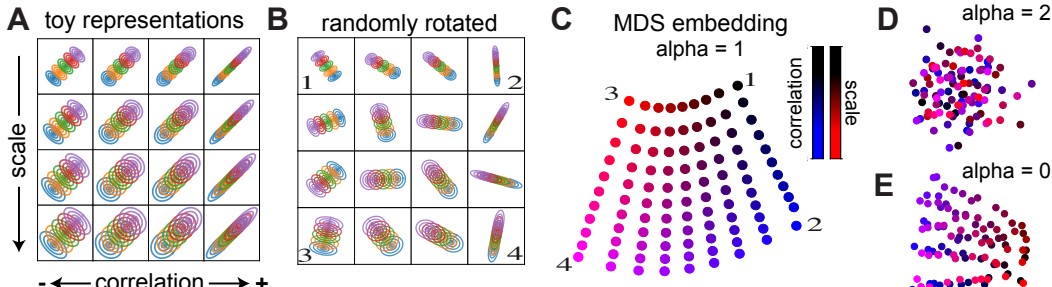

**Figure 2:** Toy Dataset. *(A)* 16 out of 99 "toy networks" with different correlation structure (horizontal axis) and covariance scale (vertical axis). Colors indicate distributions conditioned on different network inputs (as in Fig. 1B). *(B)* Same as A, with random rotations applied in neural activation space. These rotated representations are used in subsequent panels. *(C)* 2D embedding of networks in stochastic shape space ($\alpha = 1$ ground metric, $\mathcal{G} = \mathcal{O}$). Numbered points correspond to labeled representations in panel B. Colormap indicates ground truth covariance parameters. *(D)* Same as C, but with $\alpha = 2$ (covariance-insensitive). *(E)* Same as C, but with $\alpha = 0$ (mean-insensitive).

where $P_i, P_j$ are distributions with means $\boldsymbol{\mu}_i, \boldsymbol{\mu}_j$ and covariances $\boldsymbol{\Sigma}_i, \boldsymbol{\Sigma}_j$. In Appendix E we show that this defines a metric and, by extension, a shape metric when plugged into equation (5).

We can use $\alpha$ to interpolate between a Euclidean metric on the mean responses and a metric on covariances known as the Bures metric (Bhatia et al., 2019). When $\alpha = 1$ and the distributions are Gaussian, we recover the 2-Wasserstein distance. Thus, by sweeping $\alpha$, we can utilize a spectrum of stochastic shape metrics ranging from a distance that isolates differences in trial-average geometry ($\alpha = 2$) to a distance that isolates differences in noise covariance geometry ($\alpha = 0$). Distances along this spectrum can all be understood as generalizations of the usual "earth mover" interpretation of Wasserstein distance—the covariance-insensitive metric ($\alpha = 2$) only penalizes transport due to differences in the mean while the mean-insensitive metric ($\alpha = 0$) only penalizes transport due to differences in the orientation and scale of covariance. Simulation results in Supp. Figure 2 provide additional intuition for the behavior of these shape distances as $\alpha$ is adjusted between 0 to 2.

## 3 RESULTS AND APPLICATIONS

### 3.1 TOY DATASET

We begin by building intuition on a synthetic dataset in $n = 2$ dimensions with $M = 5$ inputs. Each response distribution was chosen to be Gaussian, and the mean responses were spaced linearly along the identity line. We independently varied the scale and correlation of the covariance, producing a 2D space of "toy networks." Figure 2A shows a sub-sampled $4 \times 4$ grid of toy networks. To demonstrate that stochastic shape metrics are invariant to nuisance transformations, we applied a random rotation to each network's activation space (Fig. 2B). The remaining panels show analyses for 99 randomly rotated toy networks spaced over a $11 \times 9$ grid (11 correlations and 9 scales).

Because the mean neural responses were constructed to be identical (up to rotation) across networks, existing measures of representational dissimilarity (CKA, RSA, CCA, etc.) all fail to capture the underlying structure of this toy dataset (Supp. Fig. 3). In contrast, stochastic shape distances can elegantly recover the 2D space of networks we constructed. In particular, we computed the $99 \times 99$ pairwise distance matrix between all networks (2-Wasserstein ground metric and rotation invariance, $\mathcal{G} = \mathcal{O}$) and then performing multi-dimensional scaling (MDS; Borg & Groenen, 2005) to obtain a 2D embedding. This reveals a 2D grid of networks that maps onto our constructed arrangement (Fig. 2C). Again, since the toy networks have equivalent geometries on average, a deterministic metric obtained by setting $\alpha = 2$ in eq. 7 (covariance-insensitive metric) fails to recover this structure (Fig. 2D). Setting $\alpha = 0$ in eq. 7 (mean-insensitive stochastic metric) also fails to recover a sensible 2D embedding (Fig. 2E), since covariance ellipses of opposite correlation can be aligned by a $90°$ rotation. Thus, we are only able to fully distinguish the toy networks in Figure 2A by taking *both the mean and covariance* into account when computing shape distances.

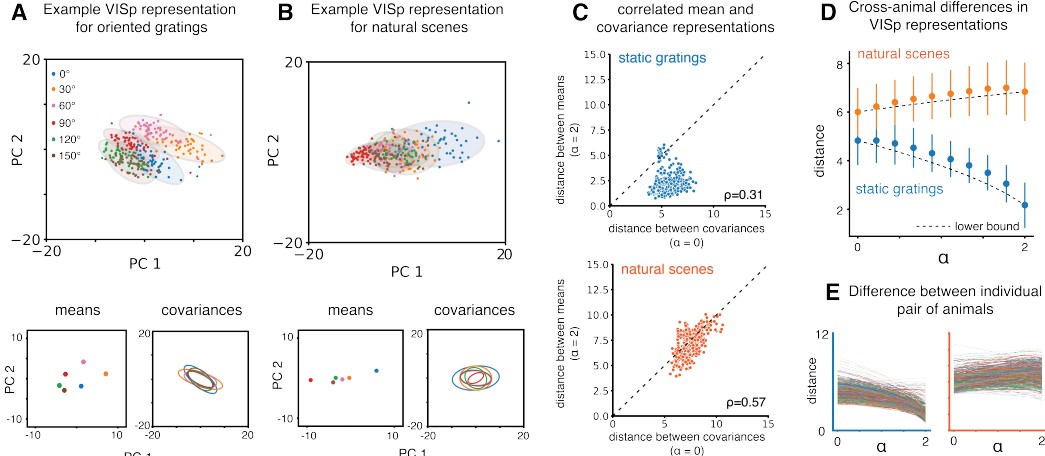

**Figure 3:** *(A)* In an example mouse VISp, neuronal responses form different means and covariances for six grating orientations. *(B)* In an example mouse VISp, neuronal responses form different means and covariances for six different natural scenes. *(C)* Covariance distances dominate differences between recording sessions for artificial grating stimuli, but not for natural scenes. *(D)* Averaged distance (across sessions) between mean responses for natural scenes is larger compared to for gratings. *(E)* Observations in *(C)* generally hold for individual pairs of recording sessions.

In Supp. Fig. 4 we show that using energy distance (eq. 4, $q = 1$) as the ground metric produces a similar result. Similar to Fig. 2C, MDS visualizations reveal the expected 2D manifold of toy networks. Indeed, these alternative distances correlate—but do not coincide exactly—with the distances shown in Figure 2 that were computed with a Wasserstein ground metric (Supp. Fig. 4D).

## 3.2 BIOLOGICAL DATA

Quantifying representational similarity is common in visual neuroscience (Shi et al., 2019; Kriegeskorte et al., 2008b). To our knowledge, past work has only quantified similarity in the geometry of trial-averaged responses and has not explored how the population geometry of noise varies across animals or brain regions (e.g. how the scale and shape of the response covariance changes). We leveraged stochastic shape metrics to perform a preliminary study on primary visual cortical recordings (VISp) from $K = 31$ mice in the Allen Brain Observatory.[1] The results we present below suggest: (a) across-animal variability in covariance geometry is comparable in magnitude to variability in trial-average geometry, (b) across-animal distances in covariance and trial-average geometry are not redundant statistics as they are only weakly correlated, and (c) the relative contributions of mean and covariance geometry to inter-animal shape distances are stimulus-dependent. Together, these results suggest that neural response distributions contain nontrivial geometric structure in their higher-order moments, and that stochastic shape metrics can help dissect this structure.

We studied population responses (evoked spike counts, see appendix B.2) to two stimulus sets: a set of 6 static oriented grating stimuli and a set of 119 natural scene images. Figure 3A shows neural responses from one animal to the oriented gratings within a principal component subspace (top), and the isolated mean and covariance geometries (bottom). Figure 3B similarly summarizes neural responses to six different natural scenes. In both cases, the scale and orientation of covariance within the first two PCs varies across stimuli. Furthermore, the scale of trial-to-trial variance was comparable to across-condition variance in the response means. These observations can be made individually within each animal, but a stochastic shape metric (2-Wasserstein ground metric, and rotation invariance $\mathcal{G} = \mathcal{O}$) enables us to quantify differences in covariance geometry *across animals*. We observed that the overall shape distance between two animals reflected a mixture of differences in trial-average and covariance geometry. Specifically, by leveraging equation (7), we observed that mean-insensitive ($\alpha = 0$) and covariance-insensitive ($\alpha = 2$) distances between animals have similar magnitudes and are weakly correlated (Fig. 3C-E).

---

[1]See appendix B.2 for full details. Data are available at: `observatory.brain-map.org/visualcoding/`

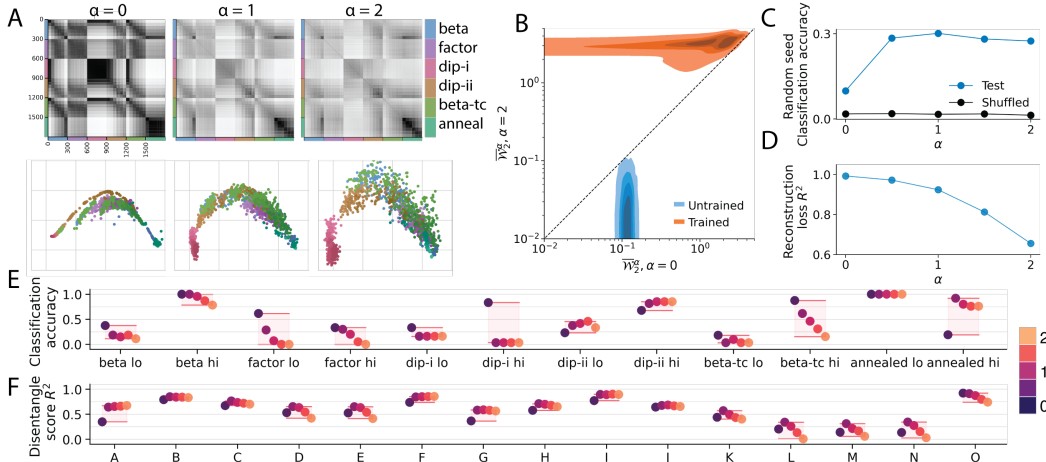

**Figure 4:** *(A)* Dissimilarity matrices with varying $\alpha$ (top) and corresponding 2D embeddings (bottom) for 1800 VAEs trained on dSprites. Six different VAE objectives (color hue) were used, each with six possible regularization strengths (tint) repeated over 50 random seeds. *(B)* Untrained networks were farther apart in mean-insensitive distance ($\alpha = 0$) while trained networks were largely separated by covariance-insensitive distance ($\alpha = 2$). *(C)* $k$NN prediction of a network's random seed. *(D)* Predicting reconstruction loss using $k$NN regression. *(E)* $k$NN prediction accuracy of VAE objective and regularization strength (hi vs lo) for metrics with different $\alpha$ (color scale). *(F)* Regression performance predicting factor disentanglement scores (see Supp. B.3 for details).

Interestingly, the ratio of these two distances was reversed across the two stimulus sets—differences in covariance geometry across animals were larger relative to differences in average for oriented gratings, while the opposite was true for natural scenes (Fig. 3C-E). Later we will show an intriguingly similar trend when comparing representations between trained and untrained deep networks.

## 3.3    VARIATIONAL AUTOENCODERS AND LATENT FACTOR DISENTANGLEMENT

Variational autoencoders (VAEs; Kingma & Welling, 2019) are a well-known class of deep generative models that map inputs, $z \in \mathcal{Z}$ (e.g. images), onto conditional latent distributions, $F(\cdot \mid z)$, which are typically parameterized as Gaussian. Thus, for each high-dimensional input $z_i$, the encoder network produces a distribution $\mathcal{N}(\mu_i, \Sigma_i)$ in a relatively low-dimensional latent space ("bottleneck layer"). Because of this, VAEs are a popular tool for unsupervised, nonlinear dimensionality reduction (Higgins et al., 2021; Seninge et al., 2021; Goffinet et al., 2021; Batty et al., 2019). However, the vast majority of papers only visualize and analyze the means, $\{\mu_1, \ldots, \mu_M\}$, and ignore the covariances, $\{\Sigma_1, \ldots, \Sigma_M\}$, generated by these models. Stochastic shape metrics enable us to compare both the mean and covariance structure learned by different VAEs. Such comparisons can help us understand how modeling choices impact learned representations (Locatello et al., 2019) and how reproducible or identifiable learned representations are in practice (Khemakhem et al., 2020).

We studied a collection of 1800 trained networks spanning six variants of the VAE framework at six regularization strengths and 50 random seeds (Locatello et al., 2019). Networks were trained on a synthetic image dataset called dSprites (Matthey et al., 2017), which is a well-established benchmark within the VAE disentanglement literature. Each image has a set of ground truth latent factors which Locatello et al. (2019) used to compute various disentanglement scores for all networks.

We computed stochastic shape distances between over 1.6 million network pairs, demonstrating the scalability of our framework. We computed rotation-invariant distances ($\mathcal{G} = \mathcal{O}$) for the generalized Wasserstein ground metric ($\alpha = 0, 0.5, 1, 1.5, 2$ in equation 7; Fig. 4A) and for energy distance ($q = 1$ in equation 4; Supp. Fig. 7A, Supp. B.3.3). In all cases, different VAE variants visibly clustered together in different regions of the stochastic shape space (Fig. 4B, Supp. Fig. 7B). Interestingly, the covariance-insensitive ($\alpha = 2$) shape distance tended to be larger than the mean-insensitive ($\alpha = 0$) shape distance (Fig. 4B), in agreement with the biological data on natural images (Fig. 3C, bottom).

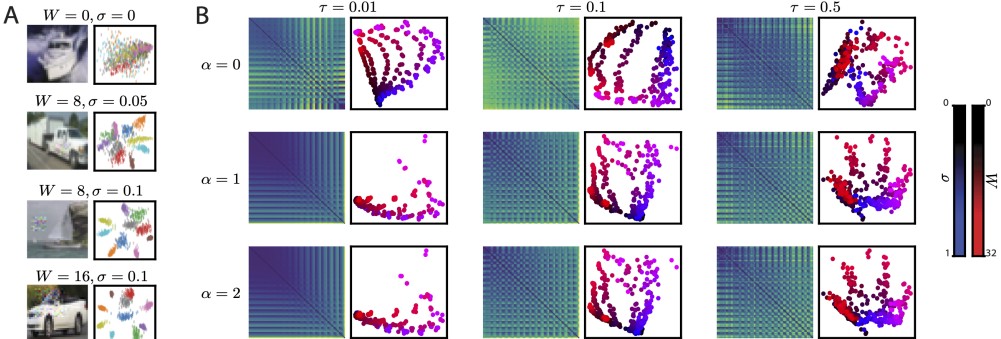

**Figure 5:** *(A) Left*, Patch-Gaussian augmented images for different patch width, $W$, and train-time noise level, $\sigma$. *Right*, MDS embedded activation vectors from Patch-Gaussian trained networks. Colors correspond to a different images, points correspond to independent samples, $\tau = 0.1$. *(B)* Distance matrices (left) and low-dimensional MDS embedding (right) of networks for different shape distances parameterized by $\alpha$, and different levels of Gaussian corruption at test time, $\tau$.

Even more interestingly, this relationship was reversed in untrained VAEs (Fig. 4B), similar to the biological data on artificial gratings (Fig. 3B, top). We trained several hundred VAEs on MNIST and CIFAR-10 to confirm these results persisted across more complex datasets (Supp. Fig. 9; Supp. B.3). Overall, this suggests that the ratio of $\alpha = 0$ and $\alpha = 2$ shape distances may be a useful summary statistic of representational complexity. We leave a detailed investigation of this to future work.

Since stochastic shape distances define proper metric spaces without triangle inequality violations, we can identify the $k$-nearest neighbors ($k$NN) of each network within this space, and use these neighborhoods to perform nonparametric classification and regression (Cover & Hart, 1967). This simple approach was sufficient to predict most characteristics of a network, including its random seed (Fig. 4C), average training reconstruction loss (Fig. 4D), its variant of the VAE objective including regularization strength (Fig. 4E), and various disentanglement scores (Fig. 4F). Detailed procedures for these analyses are provided in Appendix B.3. Notably, many of these predictions about network identity (Fig. 4E) were *more accurate* for the novel stochastic shape metrics ($0 \leq \alpha < 2$), compared to existing shape metrics ($\alpha = 2$, deterministic metric on the mean responses; Williams et al. 2021). Similarly, many disentanglement score predictions (Fig. 4F) improved when considering both covariance and means together ($0 < \alpha < 2$).

The fact that we can often infer a network's random seed from its position in stochastic shape space suggests that VAE features may have limited interpretability on this dataset. These limitations appear to apply both to the mean ($\alpha = 2$) and the covariance ($\alpha = 0$) representational geometries, as well as to intermediate interpolations. Future work that aims to assess the identifiability of VAE representations may find it useful to use stochastic shape metrics to perform similar analyses.

### 3.4  EFFECTS OF PATCH-GAUSSIAN DATA AUGMENTATION ON ARTIFICIAL DEEP NETWORKS

Despite the success of artificial neural networks on vision tasks, they are still susceptible to small input perturbations (Hendrycks & Dietterich, 2019). A simple and popular approach to induce robustness in deep networks is Patch-Gaussian augmentation (Lopes et al., 2019), which adds Gaussian noise drawn from $\mathcal{N}(0, \sigma^2)$ to random image patches of width $W$ during training (Fig. 5A, left column). At test time, network robustness is assessed with images with spatially uniform noise drawn from $\mathcal{N}(0, \tau^2)$. Importantly, the magnitude of noise at test time, $\tau$, may be distinct from noise magnitude during training, $\sigma$. From Fig. 5A (right column), we see that using Patch-Gaussian augmentation (second-fourth rows) qualitatively leads to more robust hidden layer representations on noisy data compared to networks trained without it (first row). While Patch-Gaussian augmentation is empirically successful (for quantitative details, see Lopes et al., 2019), how $W$ and $\sigma$ change hidden layer representations to confer robustness remains poorly understood.

To investigate, we trained a collection of 339 ResNet-18 networks (He et al., 2016) on CIFAR-10 (Krizhevsky, 2009), sweeping over 16 values of $W$, 7 values of $\sigma$, and 3 random seeds (see

appendix B.4 for details). While the architecture is deterministic, we can consider it to be a stochastic mapping by absorbing the random Gaussian perturbation—parameterized by $\tau$—into the first layer of the network and allowing the stochasticity to percolate through the network. Representations from a fully connected layer following the final average pooling layer were used for this analysis. We computed stochastic shape distances across all 57,291 pairs of networks across six values of $\tau$ and three shape metrics parameterized by $\alpha \in \{0, 1, 2\}$ defining the ground metric in equation (7).

Sweeping across $\alpha$ and $\tau$ revealed a rich set of relationships across these networks (Fig. 5B and Supp. Fig. 11). While a complete investigation is beyond the scope of this paper, several points are worthy of mention. First, the mean-insensitive ($\alpha = 0$, top row) and covariance-insensitive ($\alpha = 2$, bottom row) metrics produce clearly distinct MDS embeddings. Thus, the new notions of stochastic representational geometry developed in this paper (corresponding to $\alpha = 0$) provide new information to existing distance measures (corresponding to $\alpha = 2$). Second, the arrangement of networks in stochastic shape space reflects both $W$ and $\sigma$, sometimes in a 2D grid layout that maps nicely onto the hyperparameter sweep (e.g. $\alpha = 0$ and $\tau = 0.01$). Networks with the same hyperparameters but different random seeds tend to be close together in shape space. Third, the test-time noise, $\tau$, also intricately impacts the structure revealed by all metrics. Finally, embeddings based on 2-Wasserstein metric ($\alpha = 1$) qualitatively resemble embeddings based on the covariance-insensitive metric ($\alpha = 2$) rather than the mean-insensitive metric ($\alpha = 0$).

## 4  DISCUSSION AND RELATION TO PRIOR WORK

We have proposed stochastic shape metrics as a novel framework to quantify representational dissimilarity across networks that respond probabilistically to fixed inputs. Very few prior works have investigated this issue. To our knowledge, methods within the deep learning literature like CKA (Kornblith et al., 2019) have been exclusively applied to deterministic networks. Of course, the broader concept of measuring distances between probability distributions appears frequently. For example, to quantify distance between two distributions over natural images, Fréchet inception distance (FID; Heusel et al., 2017) computes the 2-Wasserstein distance within a hidden layer representation space. While FID utilizes similar concepts to our work, it addresses a very different problem—i.e., how to compare two stimulus sets within the deterministic feature space of a single neural network, rather than how to compare the feature spaces of two stochastic networks over a single stimulus set.

A select number of reports in neuroscience, particularly within the fMRI literature, have addressed how measurement noise impacts RSA (an approach very similar to CKA). Diedrichsen et al. (2020) discuss how measurement noise induces positive bias in RSA distances, and propose approaches to correct for this bias. Similarly, Cai et al. (2016) propose a Bayesian approach to RSA that performs well in low signal-to-noise regimes. These papers essentially aim to develop methods that are robust to noise, while we were motivated to directly quantify differences in noise scale and geometric structure across networks. It is also common to use Mahalanobis distances weighted by inverse noise covariance to compute intra-network representation distances (Walther et al., 2016). This procedure does not appear to quantify differences in noise structure between networks, which we verified on a simple "toy dataset" (compare Fig. 2C to Supp. Fig. 3). Furthermore, the Mahalanobis variant of RSA typically only accounts for a single, stimulus-independent noise covariance. In contrast, stochastic shape metrics account for noise statistics that change across stimuli.

Overall, our work meaningfully broadens the toolbox of representational geometry to quantify stochastic neural responses. The strengths and limitations of our work are similar to other approaches within this toolbox. A limitation in neurobiological recordings is that we only observe a subset of the total neurons in each network. Shi et al. (2019) document the effect of subsampling neurons on representational geometry. Intuitively, when the number of recorded neurons is large relative to the representational complexity, geometric features are not badly distorted (Kriegeskorte & Diedrichsen, 2016; Trautmann et al., 2019). We show that our results are not qualitatively affected by subsampling neurons in Supp. Fig. 6. Another limitation is that representational geometry does not directly shed light on the algorithmic principles of neural computation (Maheswaranathan et al., 2019). Despite these challenges, representational dissimilarity measures are one of the few quantitative tools available to compare activations across large collections of complex, black-box models, and will be a mainstay of artificial and biological network analysis for the foreseeable future.

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

# APPENDICES

## A SUPPLEMENTARY FIGURES

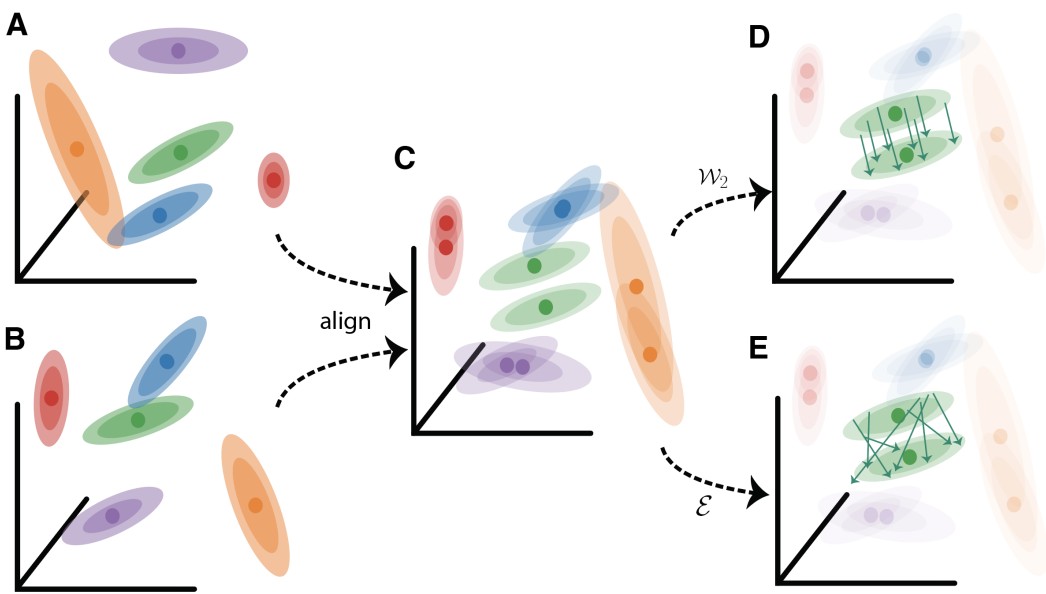

**Supplementary Figure 1:** Proposed method intuition using distances based on either $\mathcal{W}_2$ or $\mathcal{E}$ ground metrics. *(A)* and *(B)* Two example stochastic network representations to five stimuli (colors). *(C)* The optimal alignment of the representations over nuisance transformations (e.g. rotations, $\mathcal{G} = \mathcal{O}$). *(D)* Intuitively, the 2-Wasserstein distance ($\mathcal{W}_2$) is the minimum cost of turning one density (pile of dirt) to another (Villani, 2009; Peyré & Cuturi, 2019). Here we highlight the distance between the two green densities to reduce clutter. *(E)* Energy distance is based on the maximum-entropy transport map between the two distributions (Feydy et al., 2019).

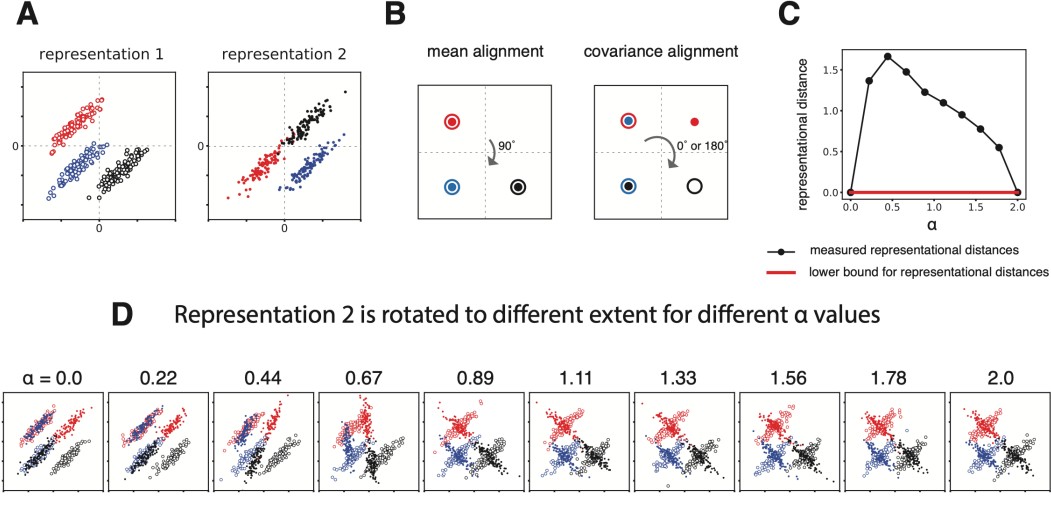

**Supplementary Figure 2:** Simulated example showing how varying the $\alpha$ parameter in $\overline{\mathcal{W}}_2^\alpha$ induces different rotational alignments between neural representations ($\mathcal{G} = \mathcal{O}$). *(A)* Two stimulated stochastic representations for three stimulus inputs. Colors represent different input conditions ($M = 3$), hollow points represent sampled representations from the first network and filled points represent sampled representations from the second network. The example is constructed so that no rotation can simultaneously align both the means and covariances. *(B)* If the stochastic metric only takes means into account ($\alpha = 2$), after rotating one of the representations by $90°$, two sets of representational means completely overlap, and the distance becomes 0. If the stochastic metric only takes covariances into account ($\alpha = 0$), the optimal alignment between the two sets of covariances is either $0°$ or $180°$, and after this rotation, distance between representations again is 0. *(C)* When both $\alpha = 0$ and $\alpha = 2$, distance between the two representations is 0, so the lower bound for the distance for $\alpha$ in the range between 0 and 2 is also 0. We computed the stochastic metric within this range of $\alpha$, and the final distance is generally above the lower bound. *(D)* Optimal rotation between the two representations at different values of $\alpha$.

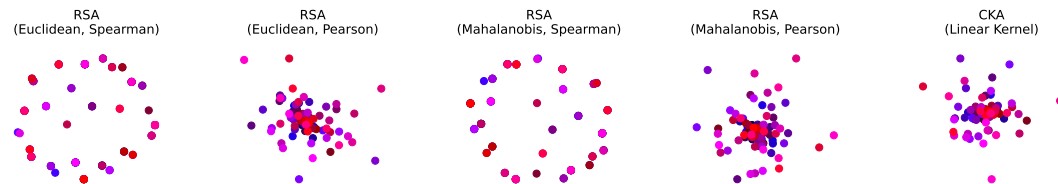

**Supplementary Figure 3:** Associated with Figure 2 from the main text. Embeddings of "toy dataset" networks (see Fig. 2A-B) visualized by multi-dimensional scaling of existing dissimilarity measures. Each point represents a network, the color scheme is the same as in Fig. 2C. All methods fail to recover a reasonable embedding which captures representational differences (compare with stochastic shape metric embeddings in Fig. 2C and Supp. Fig. 4C). Starting from the left, the first two plots use representational similarity analysis (RSA; Kriegeskorte et al. 2008a) with two forms of correlation distance (Spearman and Pearson) applied to Euclidean representational similarity matrices. The next two plots use Mahalanobis distance re-weighted by the noise covariance (Walther et al., 2016) rather than Euclidean distance. The final plot shows an embedding by centered kernel alignment with a linear filter (Kornblith et al., 2019).

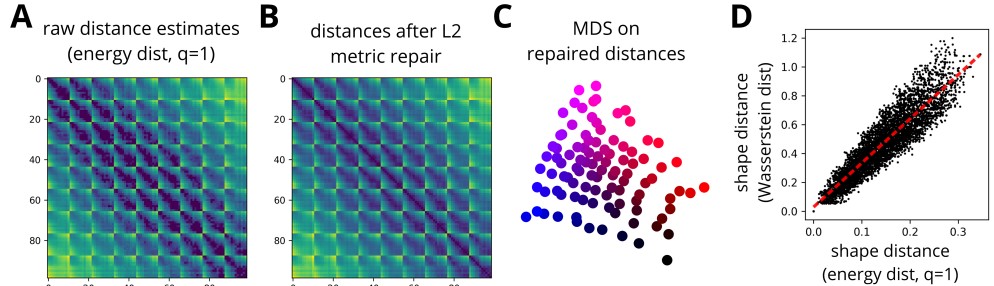

**Supplementary Figure 4:** Associated with Figure 2 from the main text. Stochastic shape metrics with energy distance also recover the "ground truth" structure of synthetic "toy data". *(A)* Matrix of estimated pairwise distances computed with $\mathcal{E}_1$ ground metric on the synthetic data shown in Fig. 2A. *(B)* Matrix of pairwise distances after quadratic metric repair (see appendix F.3) was performed to correct for minor triangle inequality violations. *(C)* Multidimensional scaling embedding of the distance matrix in panel B into 2D Euclidean space. Compare with Fig. 2C. *(D)* Linear correlation between stochastic shape distances with energy distance ground metric (i.e. off-diagonal entries of panel B) and 2-Wasserstein ground metric (i.e. off-diagonal entries of Fig. 2B). Red dashed line denotes the best linear model according to a least-squares criterion.

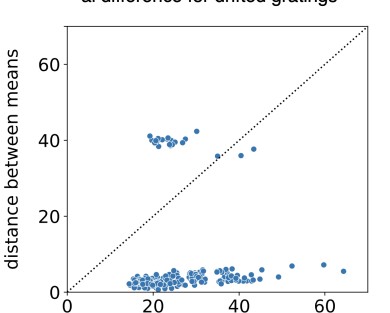

**Supplementary Figure 5:** Associated with Figure 3 from the main text. Drifted gratings (4 drifting directions, 75 repeats each) were presented in a different set of experimental sessions. Like (artificial) static gratings, representational distances across sessions for drifted gratings are dominated by covariance differences.

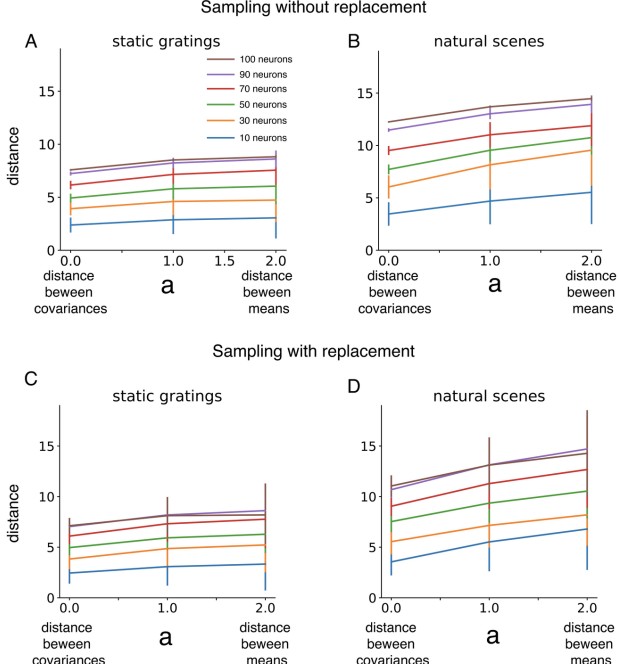

**Supplementary Figure 6:** We use a simulation to explore how the size of the neural population recording affects our conclusions about representational distances. In particular, does the ratio of mean-insensitive to covariance-insensitive across-animal distances ($\alpha = 0$ vs. $\alpha = 2$) change when we sub-sample neurons? For this simulation, we chose two mice that have 102 and 110 neurons recorded from their respective VISps. We randomly sample a subset of $n$ neurons among these recorded neurons ($n = 10, 30, 50, 70, 90, 100$), and computed representational distance ($\alpha = 0, 1, 2$) using only the subset. For panel A and B, we sampled the neurons without replacement, and for panel C and D, we sample with replacement (bootstrapping). We observe that for all tested $\alpha$, representational distance increases with the number of neurons within the subset. This is expected because distances will generically increase with the dimension (e.g. the Euclidean distance between two random vectors in a high dimensions will tend to be large, relative to low dimensions). However, the ratio of $\alpha = 0$ and $\alpha = 2$ shape distances is preserved when subsampling neurons (all lines are trending upward as a function of $\alpha$). Error bars capture how the computed distances vary across 15 random draws of $n$ neurons from the recorded population.

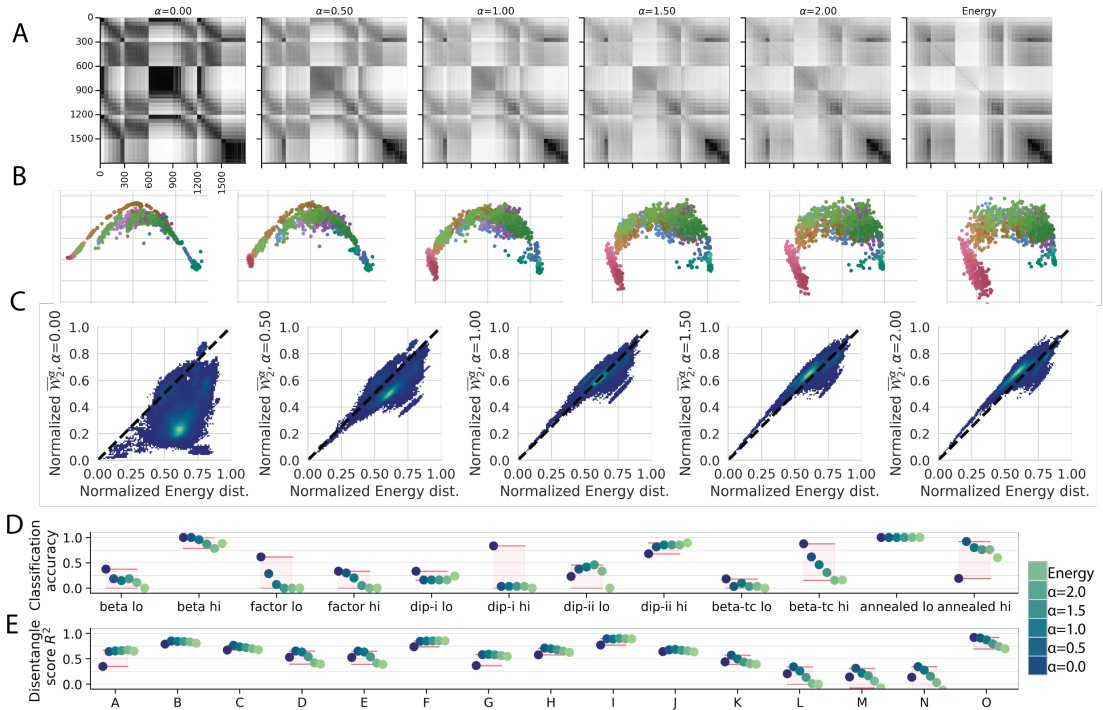

**Supplementary Figure 7:** Associated with VAE analyses (Figure 4) in the main text. *(A)* Dissimilarity matrices measured for 1800 dSprites-trained VAEs from Locatello et al. (2019) using generalized interpolated 2-Wasserstein (Equation 7) with varying $\alpha$ (first five columns), and using energy distance (Equation 4) with 64 samples for each unique input (right-most column). Row/column ordering of each matrix is the same as in Figure 4. *(B)* 2D embeddings corresponding to distance matrices in *(A)*. Colors are the same as in Figure 4. We aligned to the left-most panel using Procrustes analysis, allowing for scaling and rotations/reflections. *(C)* Re-scaling $\overline{\mathcal{W}}_2^{\alpha}$ distances and energy distances such that they lie between $[0, 1]$ reveals that the distribution of energy distances agrees best with $\overline{\mathcal{W}}_2^{\alpha=1.0}$ (middle column). *(D)* Predicting objective and regularization strength using distance matrices in *(A)*. *(E)* Predicting disentanglement scores using distance matrices in *(A)*. See Supp. B.3 for more details.

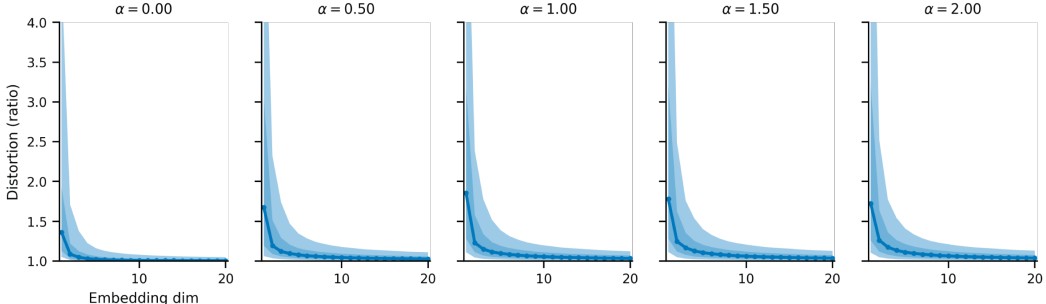

**Supplementary Figure 8:** Associated with VAE analyses (Figure 4) in the main text. Distortion induced by multidimensional scaling of $1800 \times 1800$ dissimilarity matrices with varying embedding dimensionality. Different shading represents (10th-90th) and (25th-75th) percentiles. See Supp. B.3 for details.

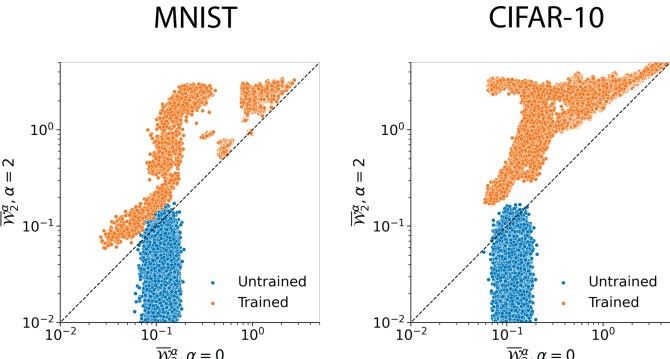

**Supplementary Figure 9:** Associated with VAE analyses (Figure 4) in the main text. We initialized 350 $\beta$-VAEs and trained them on MNIST (left) and CIFAR-10 (right) with different values of $\beta$ in the loss function. Training led to inter-network distances being dominated by covariance-insensitive ($\alpha = 2$) dissimilarity, in agreement with Figure 4B of the main text. See Supp. B.3 for training details.

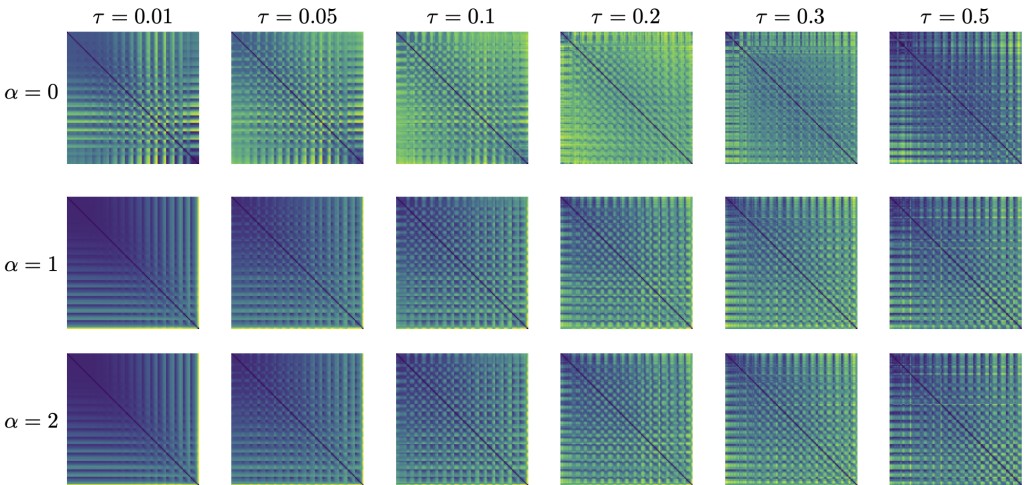

**Supplementary Figure 10:** Distance matrices for different values of $\alpha$ and $\tau$.

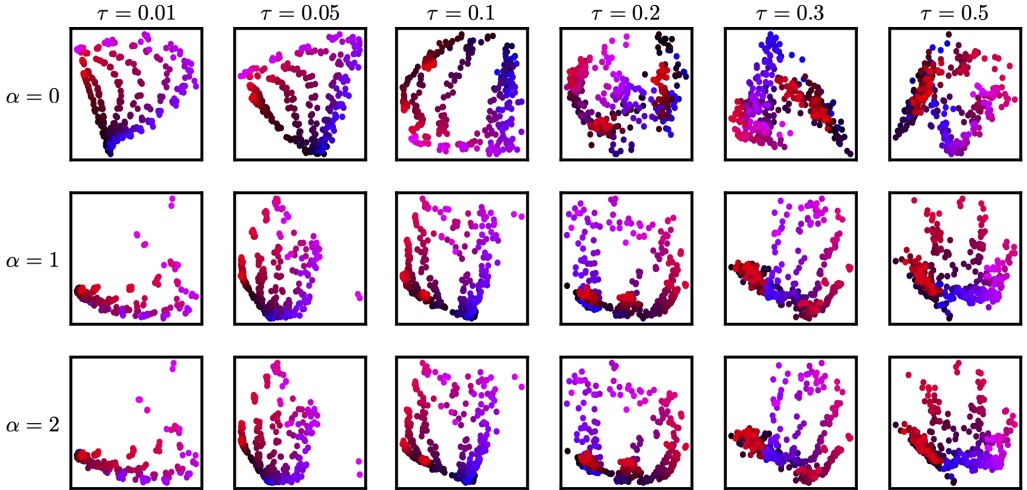

**Supplementary Figure 11:** Two dimensional embedding of the distance matrices in Fig. 10 for different values of $\alpha$ (row) and $\tau$ (column).

# B SUPPLEMENTAL METHODS

## B.1 CODE AND REPRODUCIBILITY

We have attached a zip file containing a self-contained Python script with our implementation of the interpolated generalized Wasserstein distance (equation 7) and energy distance (equation 4) in a file named `metric.py`. Additional analysis and source code will be located at `github.com/ahwillia/netrep`.

## B.2 ALLEN BRAIN OBSERVATORY DATA

### B.2.1 DATA PRE-PROCESSING.

In each recording session, gratings (6 orientations) and natural scenes (119 images) were presented to one mouse, and between 19 to 110 neurons in VISp were recorded by extracellular microelectrode arrays (Neuropixels Visual Coding dataset). Each neuron's response to an image was measured as the sum of action potentials (spikes) emitted within a 250 millisecond time window (the duration of the stimulus presentation in this dataset). To compare how similar a single set of stimuli were represented across sessions, we Gaussian-approximated the data recorded for each stimulus. Then for each stimulus class (gratings and scenes), we compared between two sets of Gaussians (one per stimulus).

Our metrics compared between sets of Gaussians that have the same dimensionality, so we performed PCA to equalize dimensionality across all sessions. We concatenated data recorded for different images within a single stimulus class, and extracted the first 19 principal components in replacement of the total number of recorded neurons for further analysis. On average, the extracted 19 principal components explained $83\%$ of the data variance in response to gratings, and $76\%$ of the variance to natural scenes.

### B.2.2 ESTIMATING RESPONSE MEAN AND COVARIANCE.

To compare between two neural representations, our stochastic metrics take two sets of Gaussian means and covariances as inputs, where each of which is estimated from the principal components (PCs) extracted from the data.

In each session, a stimulus (either a grating or a scene) was presented over 50 repeats. The number of repeats is large compared to conventional neuroscience experiments, but it is still small compared to the total number of recorded neurons (e.g. 110 neurons in one session), or the total number of PCs, which introduces challenges to covariance estimation. For the mean of each stimulus representation, we used the sample mean from the PCs. When number of samples is relatively small, sample covariance has one known bias: it tends to over-estimate large eigenvalues, and under-estimate small eigenvalues of the population covariance. One standard and effective fix in the literature is to use a shrinkage estimator ($S^*$) – a linear interpolation between an identity matrix ($I$) and the sample covariance ($S$) (e.g. Ledoit & Wolf (2004); Tong et al. (2018)):

$$S^* = \gamma I + (1 - \gamma)S. \tag{8}$$

This interpolation reduces the eigenvalue bias by balancing between eigenvalues of the sample covariance (overly skewed eigenvalue spectrum), and that of the identity matrix (flat eigenvalue spectrum). $\gamma$ of the shrinkage covariance estimator was chosen using cross-validation. To obtain the cross-validation training set, we randomly sample half of the epochs from each data trial, and for test set, we used the remaining half.

## B.3 VARIATIONAL AUTOENCODERS AND LATENT FACTOR DISENTANGLEMENT (SUPPLEMENT TO SEC. 3.3)

### B.3.1 VAE OBJECTIVES AND ARCHITECTURES USED IN THIS STUDY

Because conventional VAE encoders output a latent Gaussian conditional mean and covariance, this makes them an ideal framework with which to apply stochastic shape metrics. In particular, the interpolated Wasserstein distance (equation 7) is exact in this case. We used a set of 1800 VAEs

trained on `dSprites` from the extensive study by Locatello et al. (2019). These include six variants of the VAE objective ($\beta$, Factor, $\beta$-TC, DIP-I, DIP-II, Annealed), each with six different levels of regularization strength and 50 repetitions at different random seeds. The dimensionality of the latent representation, from which we obtained activations used in this study, was 10D. We refer the reader to their supplemental document for more details about each architecture and training scheme. The authors provided metadata associated with each network such as training hyperparameters as well as factor disentanglement scores (see below).

In addition to the VAEs trained on `dSprites`, we trained 350 $\beta$ VAEs on MNIST and CIFAR-10. To remain consistent with the study by Locatello et al. (2019), we trained $\beta$-VAEs at 6-8 different levels of regularization strength ($1 \leq \beta \leq 16$) and 50 random initialization seeds. We used the standard VAE symmetric encoder-decoder architecture with $L$ layers ($L = 3$ for MNIST and $L = 4$ for CIFAR-10), each with 64 $4 \times 4$ convolutional filters with stride 2, followed by a fully-connected layer with 256 hidden units and ReLU activations. The latent representations of these networks were diagonal Gaussians, and were 10D (MNIST) or 50D (CIFAR-10). The final 2D convolution-transpose layer of the decoder used a sigmoid nolinearity to ensure outputs were between $[0, 1]$.

We zero-padded the height and width of MNIST images from $28 \times 28 \rightarrow 32 \times 32$. Model training used batch sizes of 64 images, and up to 1000 training epochs. We used the Adam optimizer with 1E-4 learning rate and model checkpoints at each epoch. Models used in this study were from checkpoints corresponding to the lowest validation loss during training. Latent activations used for shape metric analysis in this study were obtained using a held-out test set of 3500 images.

### B.3.2 $\overline{\mathcal{W}}_2^{\alpha=0}$ VS. $\overline{\mathcal{W}}_2^{\alpha=2}$ BEFORE AND AFTER TRAINING

Fig. 4B of the main text shows that, prior to training, VAEs are primarily separated by mean-insensitive distance ($\overline{\mathcal{W}}_2^\alpha, \alpha = 0$, equation 7), whereas after training they are separated by covariance-insensitive distance ($\alpha = 2$). We sought to confirm whether this effect persisted across different datasets using VAEs trained on MNIST and CIFAR-10 (described above). SuppFig. 9 shows that pairwise network $\overline{\mathcal{W}}_2^\alpha$ distances before and after training indeed exhibit this effect on these more complex datasets. We reproduced these effects using both default PyTorch weight initialization and Kaiming weight initialization.

### B.3.3 COMPUTING ENERGY DISTANCE BETWEEN TRAINED VAEs

In addition to measuring interpolated Wasserstein distances (equation 7), we also repeated our analyses using energy distance (equation 4). Rather than requiring computing means and covariances, this method operates directly on samples. Since VAE latents are parameterized as Gaussian, we generated data by randomly sampling from the Gaussian defined by the model's conditional mean and covariance for a given input. We sampled 64 samples for 2048 images and computed pairwise energy distances between all 1800 networks in the (Locatello et al., 2019) `dSprites` dataset (SuppFig. 7). Interestingly, the energy dissimilarity matrix was qualitatively different than all of the $\overline{\mathcal{W}}_2^\alpha$ dissimilarity matrices (SuppFig. 7A). The geometry of the embedded points was accordingly different than embeddings derived from $\overline{\mathcal{W}}_2^\alpha$ distances (SuppFig. 7B).

The energy dissimilarity matrix seemed to correlate with those derived from $\overline{\mathcal{W}}_2^\alpha$ distances (Supp-Fig. 7C). We noted, however, that after re-scaling the dissimilarity matrices such that they lie between $[0, 1]$, the distribution of pairwise energy distances was most in line with interpolated Wasserstein distances when $\alpha = 1$ (SuppFig. 7C middle panel).

We repeated the classification and disentanglement $k$NN analyses done in the main text using neighborhoods defined by energy distance (SuppFig. 7D,E). In most cases, using energy distance performed as well as, but sometimes worse than $\overline{\mathcal{W}}_2^{\alpha=2}$, the covariance-insensitive Wasserstein metric. It is possible that computing energy distance using a higher number of samples per image than 64 would improve estimates and downstream regression/classification performance. In general it would be interesting to examine the effects of sample size and empirical energy distance estimate convergence. We leave a deeper investigation into this for future work.

### B.3.4 Low-dimensional projections

To determine a reasonable embedding dimensionality for $K$ networks, we performed the following analysis. Given a symmetric $K \times K$ distance matrix $D$, with elements $d(i,j)$, we used multidimensional scaling to embed $K$ networks into a low M-dimensional space. Networks in this embedded space can be encoded by a new, Euclidean distance matrix $\tilde{D}$ with elements $\tilde{d}(i,j)$. For each element on the upper-triangle of these matrices, we computed a distortion ratio,

$$\Delta(i,j) = d(i,j)/\tilde{d}(i,j) \tag{9}$$
$$\text{Distortion}(i,j) = \max(\Delta(i,j), 1/\Delta(i,j)). \tag{10}$$

By sweeping embedding dimensionality M from 1-20, we determined that using an MDS embedding dimensionality of M=15 produced reasonably minimal distortions (SuppFig. 8) for all the distance matrices. After embedding the networks into 15D, we then performed principal components analysis to obtain the scatterplots in Fig. 4A and SuppFig. 7. In the main text, we used orthogonal Procrustes to align the principal components of each subpanel to the left-most panel. For SuppFig. 7B, we again aligned all panels to the left-most panel using Procrustes analysis, but allowed for re-scaling in order to compensate for energy distances being on an arbitrary scale compared with $\overline{\mathcal{W}}_2^\alpha$ distances.

### B.3.5 VAE disentanglement metrics

For each of the 1800 VAEs trained on `dSprites`, Locatello et al. (2019) computed a large array of factor disentanglement scores proposed by previous studies. The scores abbreviated in Fig. 4F are listed in the below table. We list the different disentangement scores using the same naming convention as in the work of Locatello et al. (2019). We refer the reader to their supplement for more details on each of these scores.

|   | Disentanglement score |
|---|---|
| **A** | $\beta$-VAE eval accuracy |
| **B** | Disentanglement, Informativeness, Completeness (DCI) disentanglement |
| **C** | DCI completeness |
| **D** | DCI informativeness |
| **E** | Factor VAE eval accuracy |
| **F** | Logistic regression mean test accuracy |
| **G** | Boosted trees mean test accuracy |
| **H** | Discrete mutual information gap (MIG) |
| **I** | Modularity score |
| **J** | Explicitness test score |
| **K** | Separated Attribute Predictability (SAP) score |
| **L** | Gaussian total correlation |
| **M** | Gaussian Wasserstein correlation |
| **N** | Gaussian Wasserstein normalized correlation |
| **O** | Mutual information score |

### B.3.6 $k$-nearest neighbors analyses

Because the stochastic metrics used in this study satisfy the triangle inequality, this permitted non-parametric analyses using $k$-nearest neighbors ($k$NN) to determine whether network similarity carried information about model hyper-parameters and task performance. We used scikit-learn's `KNeighborsClassifier` and `KNeighborsRegressor` for classification and regression analyses, respectively. We withheld a test set and performed 6-fold cross-validation on the remaining data to determine $k$, the number of neighbors to use for classification/regression. We reported final performance using the average score on the held-out test set.

For classification analyses, we trained models to decode random initial seed (1/50 chance, Fig. 4C), and model objective along with regularization strength (6 objective × 6 regularization strengths in the Locatello study, i.e. 1/36 chance, Fig. 4E). In terms of regression analyses, we trained models to predict training reconstruction loss Fig. 4D and disentanglement scores Fig. 4F and reported average $R^2$ on the held-out test set.

### B.4 Additional Details for Patch-Gaussian Augmentation Experiments

#### B.4.1 Training and architecture details

We use the ResNet-18 architecture (He et al., 2016) where an intermediate fully-connected layer of dimension 100 is added after the final average pooling layer, followed by a linear readout layer. All analyses were done on the representations produced of this intermediate fully-connected layer.

Following standard practice, images were randomly cropped, followed by a random horizontal flip. A modified version of the Patch-Gaussian augmentation was applied, where the entire noisy patch is constrained to reside in the image. Lastly, we subtract off the per-channel mean and divide by the per-channel standard deviation. For the Patch-Gaussian augmentation, we swept over 16 values of patch width, $W \in \{2, 4, 6, 8, 10, 12, 14, 16, 18, 20, 22, 24, 26, 28, 30, 32\}$ and 7 different values of noise scale, $\sigma \in \{0.05, 0.1, 0.2, 0.3, 0.5, 0.8, 1.\}$, leading to $17 \times 6 = 112$ possible $(W, \sigma)$ combinations. For each $(W, \sigma)$ pair, we trained 3 networks, each with a different random seed, leading to $16 \times 7 \times 3 = 336$ networks. As a baseline, we also trained networks with no Patch-Gaussian augmentation over three random seeds, giving us 339 total networks.

We used stochastic gradient descent with a momentum of 0.9, batch size of 128 and weight decay of 1E-4. Networks were trained for 200 epochs where the learning rate was initially set to 0.1 and halved every 60 epochs.

### B.5 Visualization of Hidden Layer Representations

To visualize the effect of Patch-Gaussian hyper-parameters on hidden layer representations as shown in Fig. 5A, we randomly selected one image from each of the 10 classes, e.g. $z_1, \ldots, z_{10}$. For each image $i$—and a given value of $\tau$—we drew 100 samples from $\mathcal{N}(z_i, \tau)$ and collected the hidden layer representations, leading to 1,000 points total. Mutli-dimensional scaling was then applied to embed the representations into two dimensions.

#### B.5.1 Stochastic Shape Metric Computation

2,000 images were used for computing the stochastic shape metric. To estimate the conditional mean and covariance for each image, 1,000 samples were first drawn from $\mathcal{N}(z_i, \tau)$. The conditional mean was estimated via a Monte Carlo estimator. The conditional covariance was computed by first computing the Monte Carlo estimator and then adding 0.0001 to the diagonal to ensure the covariance is well-conditioned.

To visualize the metric shape induced by the stochastic shape metric, multi-dimensional scaling was used to embed the networks into 20 dimensions. Principal component analysis was then done to linearly project the MDS embeddings onto the top 2 principal components.

We used three different values for the interpolated Wasserstein distance, $\alpha \in \{0, 1, 2\}$ and 6 values for the magnitude of the input perturbation, $\tau \in \{0.01, 0.05, 0.1, 0.2, 0.3, 0.5\}$. All distance matrices are shown in Fig. 10. The corresponding two-dimensional embedding are shown in Fig. 11.

## C Proof of Proposition 1

We first prove two lemmas, from which the main proposition immediately follows.

**Lemma 1.** *If $\mathcal{G}$ is a group of isometries on a metric space $(d_1, S)$ then*

$$d(x, y) = \min_{\boldsymbol{T} \in \mathcal{G}} d_1(x, \boldsymbol{T}(y)) \tag{11}$$

*is a pseudometric which can be used to define a metric over equivalence classes $[x] = \{y \mid y \sim x\}$ where the equivalence relation is defined as:*

$$x \sim y \iff \exists\, \boldsymbol{T} \in \mathcal{G} \quad \text{such that} \quad x = \boldsymbol{T}(y) \tag{12}$$

*Proof.* This proof is more or less reproduced from Williams et al. (2021), and similar arguments can be found elsewhere within the statistical shape analysis literature.

The equivalence relation in equation (12) is self-evident. This simply states that $d(x, y) = 0$ if and only if $x = \boldsymbol{T}(y)$ for some alignment transformation $\boldsymbol{T} \in \mathcal{G}$. Then we define our equivalence relation as: $x \sim y$ if and only if $d(x, y) = 0$. In other words, although $d$ technically only defines a pseudometric on $S$, it is easily associated to a proper metric on a set of equivalence classes, i.e. the quotient space $(S/\sim)$. See Howes (1995) for more background details (page 27, in particular).

Now we prove that $d$ is symmetric. Let $\boldsymbol{T}_{xy}$ denote the optimal transformation from $Y$ to $X$. That is, $\boldsymbol{T}_{xy} = \operatorname{argmin}_{\boldsymbol{T} \in \mathcal{G}} d_1(x, \boldsymbol{T}(y))$ and $\boldsymbol{T}_{yx} = \operatorname{argmin}_{\boldsymbol{T} \in \mathcal{G}} d_1(y, \boldsymbol{T}(x))$. Then, using the fact that $d_1$ is symmetric and that $\mathcal{G}$ defines a group of isometries, we have

$$d(x, y) = d_1(x, \boldsymbol{T}_{xy}(y)) = d_1(\boldsymbol{T}_{xy}(y), x) = d_1(y, \boldsymbol{T}_{xy}^{-1}(x))) \le d_1(y, \boldsymbol{T}_{yx}(x))) = d(y, x)$$

but also

$$d(y, x) = d_1(y, \boldsymbol{T}_{yx}(x)) = d_1(\boldsymbol{T}_{yx}(x), y) = d_1(x, \boldsymbol{T}_{yx}^{-1}(y))) \le d_1(x, \boldsymbol{T}_{xy}(y))) = d(x, y).$$

The only way for both inequalities to hold is for $d(x, y) = d(y, x)$. Also, we see that $\boldsymbol{T}_{xy} = \boldsymbol{T}_{yx}^{-1}$, which we will exploit below.

It remains to prove the triangle inequality. This is done as follows:

$$d(x, y) = d_1(x, \boldsymbol{T}_{xy}(y)) \tag{13}$$
$$\le d_1(x, \boldsymbol{T}_{xz}(\boldsymbol{T}_{zy}(y))) \tag{14}$$
$$\le d_1(x, \boldsymbol{T}_{xz}(z)) + d_1(\boldsymbol{T}_{xz}(z), \boldsymbol{T}_{xz}(\boldsymbol{T}_{zy}(y))) \tag{15}$$
$$= d_1(x, \boldsymbol{T}_{xz}(z)) + d_1(Z, \boldsymbol{T}_{zy}(y)) \tag{16}$$
$$= d(x, z) + d(z, y) \tag{17}$$

The first inequality follows from replacing the optimal alignment, $\boldsymbol{T}_{xy}$, with a sub-optimal alignment, given by function composition $\boldsymbol{T}_{xz} \circ \boldsymbol{T}_{zy}$. (Recall that $\mathcal{G}$ is a group and so is closed under function compositions.) The second inequality follows from the triangle inequality on $d_1$, after choosing $\boldsymbol{T}_{xz}(z)$ as the midpoint. The penultimate step follows from $\boldsymbol{T}_{xz}^{-1}$ being an isometry on $d_1$ and since $\boldsymbol{T}_{xz} \in \mathcal{G}$, we have $\boldsymbol{T}_{xz}^{-1} \in \mathcal{G}$ by the group properties of $\mathcal{G}$.

$\square$

**Lemma 2.** *Let $(d_2, S_2)$ be a metric space, let $f(\cdot)$ and $g(\cdot)$ be functions mapping $\mathcal{Z} \mapsto S_2$, and let $Q$ be a probability distribution supported on $\mathcal{Z}$. Then,*

$$d_1(f, g) = \Big( \mathop{\mathbb{E}}_{z \sim Q} d_2^2(f(z), g(z)) \Big)^{1/2} \tag{18}$$

*is a metric over the set of functions mapping $\mathcal{Z} \mapsto S_2$.*

*Proof.* Since $d_2$ is a metric, we have $d_2(x, y) > 0$ if $x \ne y$. Recall our assumption that the support of $Q$ equals $\mathcal{Z}$. Thus, if there exists a $z \in \mathcal{Z}$ for which $f(z) \ne g(z)$, the expectation will evaluate to a positive number and we have $d_1(f, g) > 0$. So we conclude $d_1(f, g) = 0$ if and only if $f$ and $g$ define the exact same mapping from $\mathcal{Z} \mapsto S_2$.

It is also obvious that $d_1(f, g) = d_1(g, f)$, due to the symmetry of $d_2$. Thus, it only remains to prove the triangle inequality.

Fix any function $h : \mathcal{Z} \mapsto S$. Due to the triangle inequality on $d_2$, we have:

$$d_1(f, g) = \Big( \mathop{\mathbb{E}}_{z \sim Q} d_2^2(f(z), g(z)) \Big)^{1/2} \le \Big( \mathop{\mathbb{E}}_{z \sim Q} \big( d_2(f(z), h(z)) + d_2(h(z), g(z)) \big)^2 \Big)^{1/2} \tag{19}$$

Now let $X = d_2(f(z), h(z))$ and $Y = d_2(h(z), g(z))$. Note that $z$ is a random variable (sampled from $Q$), and so $X$ and $Y$ are also random variables. We now recall two elementary facts: $\|X\|_2 = (\mathbb{E}[X^2])^{1/2}$ defines a norm over random variables, and $\|X + Y\|_2 \le \|X\|_2 + \|Y\|_2$ for any two random variables (Minkowski's inequality). Our definitions of $X$ and $Y$ imply that the right hand side of equation (19) can be re-written as $\|X + Y\|_2$. And we can therefore conclude the proof since:

$$d_1(f, g) \le \|X + Y\|_2 \le \|X\|_2 + \|Y\|_2 = d_1(f, h) + d_1(h, g). \tag{20}$$

$\square$

**Main proof.** Let us restate and then prove proposition 1. We want to show that the following:

$$d(F_i, F_j) = \min_{\boldsymbol{T} \in \mathcal{G}} \left( \mathbb{E}_{\boldsymbol{z} \sim Q} \left[ \mathcal{D}^2 \left( F_i^{\phi_i}(\cdot \mid \boldsymbol{z}), F_j^{\phi_j}(\cdot \mid \boldsymbol{z}) \circ \boldsymbol{T}^{-1} \right) \right] \right)^{1/2} \tag{21}$$

is a pseudometric over stochastic networks—i.e, a pseudometric over functions $F$ that map inputs $\boldsymbol{z} \in \mathcal{Z}$ onto probability distributions. Recall that $F^\phi(\cdot \mid \boldsymbol{z})$ is a shorthand notation for $F(\phi^{-1}(\cdot) \mid \boldsymbol{z})$ where $\phi^{-1}$ is the pre-image of $\phi$.

Our key assumptions are that $\mathcal{D}(\cdot, \cdot)$ is a metric over probability distributions and that $\mathcal{G}$ is a group of isometry transformations with respect to this metric—i.e., for any pair of probability distributions $F$ and $G$, we have that:

$$\mathcal{D}(F, G) = \mathcal{D}(F \circ \boldsymbol{T}^{-1}, G \circ \boldsymbol{T}^{-1}) \tag{22}$$

for any $\boldsymbol{T} \in \mathcal{G}$. It is well-known that the Wasserstein distance (Villani, 2009) and energy distance (Sejdinovic et al., 2013; Székely & Rizzo, 2017) are probability metrics. Further, it is easy to show that orthogonal pushforward transformations are isometries for both metrics. For example, we have for the 2-Wasserstein distance that:

$$\mathcal{W}_2^2(P, Q) = \inf \mathbb{E} \|X - Y\|^2 = \inf \mathbb{E} \|\boldsymbol{T}X - \boldsymbol{T}Y\|^2 = \mathcal{W}_2^2(P \circ \boldsymbol{T}^{-1}, Q \circ \boldsymbol{T}^{-1}) \tag{23}$$

for any orthogonal transformation $\boldsymbol{T}$. Thus, for our purposes we can think of $\mathcal{G}$ as being any subgroup of the orthogonal group.

Now that we have reminded ourselves of the main proposition, let us turn to the proof.

*Proof.* Let us define:

$$d_1(F_i^\phi, F_j^\phi) = \left( \mathbb{E}_{\boldsymbol{z} \sim Q} \left[ \mathcal{D}^2 \left( F_i^\phi(\cdot \mid \boldsymbol{z}), F_j^\phi(\cdot \mid \boldsymbol{z}) \right) \right] \right)^{1/2}. \tag{24}$$

Plugging this into equation (21), we have:

$$d(F_i, F_j) = \min_{\boldsymbol{T} \in \mathcal{G}} d_1(F_i^\phi, F_j^\phi \circ \boldsymbol{T}^{-1}). \tag{25}$$

Lemma 2 tells us that $d_1$ is a metric. Thus, lemma 1 applies to equation (25). This permits us to conclude that $d$ is a pseudometric and defines a metric over sets of equivalent neural representations, as claimed. $\qquad \square$

## D   PRACTICAL ESTIMATION OF STOCHASTIC SHAPE METRICS

In both biological and artificial networks, we do not have parametric forms for the conditional distributions over neural population responses. Instead, we can only draw samples from these distributions—e.g., by feeding an input into an artificial network and performing a stochastic forward pass, or by recording evoked spike counts to a sensory stimulus in biological data. We consider a simple experimental setup: we are given $K$ stochastic neural networks $\{F_1, \dots, F_K\}$, $M$ network inputs or conditions $\{\boldsymbol{z}_1, \dots, \boldsymbol{z}_M\}$, and $L$ repeated observations or measurements of the neural responses to each input. For example, in an artificial network that ingests image data, $M$ would denote the number of images in a test set and $L$ denotes the number of samples per image. Let $\boldsymbol{x}_\ell^{(km)} \in \mathbb{R}^n$ to denote sample $\ell$, from network $k$, to condition $m$. That is,

$$\boldsymbol{x}_\ell^{(km)} \sim F_k^\phi(\boldsymbol{x} \mid \boldsymbol{z}_m) \quad \text{i.i.d. for } (\ell, m, k) \in \{1, \dots, L\} \times \{1, \dots, M\} \times \{1, \dots, K\}. \tag{26}$$

### D.1   METRICS BASED ON 2-WASSERSTEIN DISTANCE AND GAUSSIAN ASSUMPTION

Our main assumption in this section is that distributions over neural activations are multivariate Gaussians. That is, for each stochastic network and every input $\boldsymbol{z} \in \mathcal{Z}$, we have $F_i^\phi(\boldsymbol{z}) = \mathcal{N}(\boldsymbol{\mu}_i(\boldsymbol{z}), \boldsymbol{\Sigma}_i(\boldsymbol{z}))$, where $\boldsymbol{\mu}_i : \mathcal{Z} \mapsto \mathbb{R}^n$ and $\boldsymbol{\Sigma}_i : \mathcal{Z} \mapsto \mathbb{S}^{n \times n}$. If $\boldsymbol{T} : \mathbb{R}^n \mapsto \mathbb{R}^n$ is a linear pushforward map, then the pushforward measure is still Gaussian and is defined by $F_j^\phi(\boldsymbol{z}) \circ \boldsymbol{T}^{-1} = \mathcal{N}(\boldsymbol{T}\boldsymbol{\mu}_j(\boldsymbol{z}), \boldsymbol{T}\boldsymbol{\Sigma}_j(\boldsymbol{z})\boldsymbol{T}^\top)$.

The 2-Wasserstein distance between two multivariate Gaussian distributions has a well-known closed form expression (Peyré & Cuturi, 2019, Remark 2.31):

$$\mathcal{W}_2\big(\mathcal{N}(\boldsymbol{\mu}_i, \boldsymbol{\Sigma}_i), \mathcal{N}(\boldsymbol{\mu}_j, \boldsymbol{\Sigma}_j)\big) = \big(\|\boldsymbol{\mu}_i - \boldsymbol{\mu}_j\|^2 + \mathcal{B}(\boldsymbol{\Sigma}_i, \boldsymbol{\Sigma}_j)^2\big)^{1/2} \tag{27}$$

where $\mathcal{B}(\cdot, \cdot)$ is the Bures metric between positive definite matrices. Typically one sees the Bures metric defined as:

$$\mathcal{B}(\boldsymbol{\Sigma}_i, \boldsymbol{\Sigma}_j) = \Big(\text{Tr}[\boldsymbol{\Sigma}_i] + \text{Tr}[\boldsymbol{\Sigma}_j] - 2\,\text{Tr}[\boldsymbol{\Sigma}_i^{1/2}\boldsymbol{\Sigma}_j\boldsymbol{\Sigma}_i^{1/2}]^{1/2}\Big)^{1/2}. \tag{28}$$

If we use this expression, minimizing $\mathcal{B}(\boldsymbol{\Sigma}_i, \boldsymbol{T}\boldsymbol{\Sigma}_j\boldsymbol{T}^\top)$ over nuisance transformations $\boldsymbol{T} \in \mathcal{G}$ is not straightforward.[2] However, an equivalent formulation of the Bures metric is:

$$\mathcal{B}(\boldsymbol{\Sigma}_i, \boldsymbol{\Sigma}_j) = \min_{\boldsymbol{U}} \|\boldsymbol{\Sigma}_i^{1/2} - \boldsymbol{\Sigma}_j^{1/2}\boldsymbol{U}\|_F \tag{29}$$

where the minimization is over $\boldsymbol{U} \in \mathcal{O}(n)$. The equivalence between equations (28) and (29) is already established in the literature (see Theorem 1 of Bhatia et al. 2019). For the sake of completeness we have included a proof in appendix F.4.

Recall that we are given sampled neural responses $\{\boldsymbol{x}_\ell^{(km)}\}_{k,m,\ell}^{K,M,L}$ as specified in equation (26). Using these, we can estimate the mean and covariance of each distribution:

$$\hat{\boldsymbol{\mu}}_k(\boldsymbol{z}_m) = \frac{1}{L}\sum_\ell \boldsymbol{x}_\ell^{(km)} \quad \text{and} \quad \hat{\boldsymbol{\Sigma}}_k(\boldsymbol{z}_m) = \frac{1}{L}\sum_\ell \boldsymbol{x}_\ell^{(km)}\boldsymbol{x}_\ell^{(km)\top} - \hat{\boldsymbol{\mu}}_k(\boldsymbol{z}_m)\hat{\boldsymbol{\mu}}_k(\boldsymbol{z}_m)^\top. \tag{30}$$

Here, we've used the typical maximum likelihood estimators. However, any consistent estimator will suffice.

The proposition below summarizes the main result of this section. Using this proposition, we produce an estimate of the distance between two stochastic networks by alternating minimization (i.e. block coordinate descent) over $\boldsymbol{T}, \boldsymbol{U}_1, \dots, \boldsymbol{U}_M$. Each parameter update can often be solved exactly. For example, we typically consider the case of orthogonal nuisance transformations, i.e. $\mathcal{G} = \mathcal{O}(n)$, in which case all parameter updates correspond to solving an orthogonal Procrustes problem (Gower & Dijksterhuis, 2004). Further, the minimizations over $\{\boldsymbol{U}_1, \dots, \boldsymbol{U}_M\}$ can be done in parallel.

**Proposition 2.** *If $F_i^\phi(\boldsymbol{z})$ and $F_j^\phi(\boldsymbol{z})$ are both Gaussian for all $\boldsymbol{z} \in \mathcal{Z}$, then:*

$$\hat{d}(F_i, F_j) = \min_{\boldsymbol{T}, \boldsymbol{U}_1, \dots, \boldsymbol{U}_M} \Big(\frac{1}{M}\sum_{m=1}^M \|\hat{\boldsymbol{\mu}}_i(\boldsymbol{z}_m) - \boldsymbol{T}\hat{\boldsymbol{\mu}}_j(\boldsymbol{z}_m)\|^2 + \|\hat{\boldsymbol{\Sigma}}_i(\boldsymbol{z}_m)^{1/2} - \boldsymbol{T}\hat{\boldsymbol{\Sigma}}_j(\boldsymbol{z}_m)^{1/2}\boldsymbol{U}_m\|_F^2\Big)^{1/2}$$

*is a consistent estimator of a stochastic shape distance (eq. 5) with the 2-Wasserstein distance used as the "ground metric." The minimization in the above equation is performed over $\boldsymbol{T} \in \mathcal{G}$ and $\boldsymbol{U}_m \in \mathcal{O}(n)$ for all $m \in \{1, \dots, M\}$.*

*Proof.* Plugging equations (27) and (29) into our definition of stochastic distance (eq. 5 from proposition 1), we have:

$$d(F_i, F_j) = \min_{\boldsymbol{T}} \Big(\mathbb{E}_{\boldsymbol{z}\sim Q}\|\boldsymbol{\mu}_i(\boldsymbol{z}) - \boldsymbol{T}\boldsymbol{\mu}_j(\boldsymbol{z})\|^2 + \min_{\boldsymbol{U}} \|\boldsymbol{\Sigma}_i(\boldsymbol{z})^{1/2} - \boldsymbol{T}\boldsymbol{\Sigma}_j(\boldsymbol{z})^{1/2}\boldsymbol{T}^\top\boldsymbol{U}\|_F^2\Big)^{1/2}. \tag{31}$$

Given $M$ i.i.d. samples $\boldsymbol{z}_m \sim Q$ for $m \in \{1, \dots, M\}$, we can estimate the expectation with an empirical average:

$$\min_{\boldsymbol{T}} \Big(\frac{1}{M}\sum_{m=1}^M \|\boldsymbol{\mu}_i(\boldsymbol{z}_m) - \boldsymbol{T}\boldsymbol{\mu}_j(\boldsymbol{z}_m)\|^2 + \min_{\widetilde{\boldsymbol{U}}_m} \|\boldsymbol{\Sigma}_i(\boldsymbol{z}_m)^{1/2} - \boldsymbol{T}\boldsymbol{\Sigma}_j(\boldsymbol{z}_m)^{1/2}\boldsymbol{T}^\top\widetilde{\boldsymbol{U}}_m\|_F^2\Big)^{1/2} \tag{32}$$

---

[2]Although there are certain tricks one can exploit to compute the gradient (Newton-Schulz iterations), the constraint that $\boldsymbol{T} \in \mathcal{G}$ is non-trivial. When $\mathcal{G}$ is a continuous manifold (e.g. the orthogonal or special orthogonal group), one can resort to manifold optimization algorithms. These algorithms are somewhat cumbersome but nonetheless a plausible approach. However, even this would not cover the case where $\mathcal{G}$ is a discrete set (e.g., the permutation group).

Next, we pull out the minimization over each $\widetilde{U}_m \in \mathcal{O}(n)$ outside the sum. Additionally, since $\mathcal{G}$ is a group of isometries on $\mathbb{R}^n$, we know that $\mathcal{G}$ is a subgroup of the orthogonal group. Thus, every $T \in \mathcal{G}$ is an orthogonal matrix, so $T^\top \widetilde{U}_m$ is also an orthogonal matrix. Thus we introduce a change of variables $U_m = T^\top \widetilde{U}_m$ and minimize over $U_m \in \mathcal{O}(n)$, as this attains the same minimum value. In summary, we have:

$$\min_{T, U_1,\ldots,U_M} \left( \tfrac{1}{M} \sum_{m=1}^M \|\boldsymbol{\mu}_i(\boldsymbol{z}_m) - T\boldsymbol{\mu}_j(\boldsymbol{z}_m)\|^2 + \|\boldsymbol{\Sigma}_i(\boldsymbol{z}_m)^{1/2} - T\boldsymbol{\Sigma}_j(\boldsymbol{z}_m)^{1/2}U_m\|_F^2 \right)^{1/2}. \quad (33)$$

The only remaining step is to replace every $\boldsymbol{\mu}(\boldsymbol{z})$ and $\boldsymbol{\Sigma}(\boldsymbol{z})$ with some consistent estimator, such as the empirical mean and covariance (see eq. 30). In the limit as $M \to \infty$ and $L \to \infty$, we have convergence to the true distance due to the law of large numbers. $\qquad\square$

### D.1.1 Algorithmic complexity and computational considerations

To compute distances using the 2-Wasserstein ground metric between two Gaussian-distributed stochastic network representations (Eq. 6), we used closed form updates of the orthogonal procrustes problem (Gower & Dijksterhuis, 2004) for $T \in \mathcal{O}$ and $\{U_m\}_{m=1}^M$ in alternation, using $S$ iterations. Importantly, $T$ and each $U_m$ can be solved exactly at each alternation (described above). For $K$ stochastic networks, we must consider $\mathcal{O}(K^2)$ total pairwise comparisons.

Computing the optimal $T$ at each step involves aligning two stochastic representations, each comprising a stacked matrix of $M$ $n$-dimensional means and $M$ $n \times n$ covariances using Procrustes alignment (Gower & Dijksterhuis, 2004). This involves a matrix multiplication and singular value decomposition with $\mathcal{O}(Mn^3)$ combined complexity, assuming $n < M$. Similarly, at each step, computing the Bures metric requires solving for $M$ $n \times n$ orthogonal matrices, $\{U_m\}_{m=1}^M$, with total complexity $\mathcal{O}(Mn^3)$. Thus, the total algorithmic worst-case time complexity is $\mathcal{O}(K^2 S M n^3)$.

Notably, this computation is highly parallelizable over the $K^2$ pairwise comparisons. For the VAE and patch-Gaussian results in the main text (Figures 4 and 5), we distributed the distance matrix calculation by distributing pairwise comparisons over single CPU cores, with each comparison taking a few seconds to complete.

### D.2 Metrics based on Energy distance

This section outlines an alternative measure of stochastic representational distance that does not require any parametric assumption (e.g. Gaussian) on stochastic neural responses. We use *energy distance*, $\mathcal{E}_q$ defined in equation (4), as the ground metric appearing in proposition 1. As explained in the main text, this distance has favorable estimation properties in high dimensional spaces relative to the Wasserstein distances. It remains an open problem to develop estimation procedures for Wasserstein-based stochastic shape distances without the assumption of Gaussianity.

To compute the stochastic shape distance, we need to solve the following optimization problem:

$$\operatorname*{argmin}_{T \in \mathcal{G}} \mathbb{E}_{z \sim Q} \left[ \mathcal{E}_q^2 \left( F_i^\phi(\boldsymbol{z}), F_j^\phi(\boldsymbol{z}) \circ T^{-1} \right) \right] \quad (34)$$

Note that we have squared the expression occurring in the main proposition—i.e., we have dropped the $(\cdot)^{1/2}$ operation—as this does not effect the optimal alignment transformation.

First, let's focus on the innermost term inside the expectation of equation (34). Let $X_i, X_i' \sim F_i^\phi(\cdot \mid \boldsymbol{z})$ and $X_j, X_j' \sim F_j^\phi(\cdot \mid \boldsymbol{z})$, independently and treating the input $\boldsymbol{z}$ as fixed for now. Now, since $\mathcal{G}$ is a group of isometries with respect to the Euclidean norm, we have:

$$\mathcal{E}_q^2(F_i^\phi(\cdot \mid \boldsymbol{z}), F_j^\phi(\cdot \mid \boldsymbol{z}) \circ T^{-1}) = \mathbb{E}\|X_i - TX_j\|^q - \tfrac{1}{2}\mathbb{E}\|X_i - X_i'\|^q - \tfrac{1}{2}\mathbb{E}\|TX_j - TX_j'\|^q$$

$$= \mathbb{E}\|X_i - TX_j\|^q - \tfrac{1}{2}\mathbb{E}\|X_i - X_i'\|^q - \tfrac{1}{2}\mathbb{E}\|X_j - X_j'\|^q$$

The final two terms are constant with respect to $T$, so we can drop them from the objective function without effecting the result. Thus, equation (34) can be simplified to:

$$\operatorname*{argmin}_{T \in \mathcal{G}} \mathbb{E}_{z \sim Q} \left[ \mathbb{E}\|X_i - TX_j\|^q \right] \quad (35)$$

Here, the outer expectation is over network inputs $\boldsymbol{z}$ and the inner expectation is over conditional distributions, $X_i \sim F_i^\phi(\cdot \mid \boldsymbol{z})$ and $X_j \sim F_j^\phi(\cdot \mid \boldsymbol{z})$.

Recall again that we are given sampled responses $\{\boldsymbol{x}_\ell^{(km)}\}_{k,m,\ell}^{K,M,L}$ as specified in equation (26). To construct a consistent estimator, we evoke the law of large numbers to replace the expectations with empirical averages. Equation (35) becomes:

$$\operatorname*{argmin}_{\boldsymbol{T} \in \mathcal{G}} \frac{1}{ML^2} \sum_{m=1}^M \sum_{\ell=1}^L \sum_{p=1}^L \|\boldsymbol{x}_\ell^{(im)} - \boldsymbol{T}\boldsymbol{x}_p^{(jm)}\|^q \tag{36}$$

When $q = 2$, the optimal $\boldsymbol{T} \in \mathcal{G}$ can often be identified efficiently—e.g., by solving a Procrustes problem when $\mathcal{G} = \mathcal{O}$ (Gower & Dijksterhuis, 2004) or a linear assignment problem when $\mathcal{G} = \mathcal{P}$ (Burkard et al., 2012). However, when $q = 2$ the stochastic shape distance only depends on the mean and is insensitive to higher-order moments of the neural response (see appendix F.1) In the more interesting case where $q \neq 2$, we can use iteratively re-weighted least squares (Kuhn, 1973) to identify the solution.[3] Details of this well-known algorithm are provided in appendix F.2.

Now, let $\boldsymbol{T}^* \in \mathcal{G}$ be the solution to equation (36). Using this it is straightforward to estimate the desired stochastic shape distance. Our estimate of $d(F_i, F_j)$ is:

$$\frac{1}{m} \sum_m \left( \frac{1}{L^2} \sum_{\ell,p} \|\boldsymbol{x}_\ell^{(im)} - \boldsymbol{T}^* \boldsymbol{x}_p^{(jm)}\|^q - \frac{1}{L(L-1)} \sum_{\ell>p} \|\boldsymbol{x}_\ell^{(im)} - \boldsymbol{x}_p^{(im)}\|^q - \frac{1}{L(L-1)} \sum_{\ell>p} \|\boldsymbol{x}_\ell^{(jm)} - \boldsymbol{x}_p^{(jm)}\|^q \right)$$

where the sums over $\ell > p$ are over all $L(L - 1)/2$ pairwise combinations between $L$ sampled activations. Each of the three terms in the expression above is a consistent (though not unbiased) estimator of its corresponding term in definition of energy distance (eq. 4). However, if the final two terms above are over-estimated in magnitude and the first term is under-estimated, the overall estimate of $d(F_i, F_j)$ may be negative. This violates perhaps the most important property of a metric space that distances should be nonnegative. Triangle inequality violations are also possible.

We propose a simple fix using basic ideas from the literature on *metric repair* (Brickell et al., 2008). Given a collection of $K$ networks, we use the procedure above to compute an estimate of the $K \times K$ distance matrix $\widetilde{\boldsymbol{D}}$ where $\widetilde{\boldsymbol{D}}_{ij} \approx d(F_i, F_j)$. We then find the matrix $\boldsymbol{D}^*$ that is closest to our estimate $\widetilde{\boldsymbol{D}}$ according a quadratic loss, and which satisfies all the axioms of a metric space. This amounts to solving a quadratic program, as detailed in appendix F.3.

## E INTERPOLATED METRICS BETWEEN MEAN-SENSITIVE AND COVARIANCE-SENSITIVE DISTANCES

### E.1 PROOF THAT $\overline{\mathcal{W}}_2^\alpha$ IS A METRIC

We start by proving a well-known and basic lemma, which states that the $\ell_p$ norm of a collection of metrics also defines a metric.

**Lemma 3.** *Let $d_1, \ldots, d_n$ be a collection of metrics on a set $\mathcal{S}$. Then, for any $p > 1$,*

$$d(x, y) = \sqrt[p]{d_1(x,y)^p + \ldots + d_n(x,y)^p} \tag{37}$$

*is a metric on $\mathcal{S}$.*

*Proof.* It is obvious that $d(x, y) = 0$ if and only if $d_1(x, y) = \ldots = d_n(x, y) = 0$ and that $d(x, y) = d(y, x)$. So it is only non-trivial to prove the triangle inequality.

Let $\boldsymbol{d}(x, y)$ denote the vector in $\mathbb{R}^n$ holding each distance. That is:

$$\boldsymbol{d}(x, y) = \begin{bmatrix} d_1(x, y) \\ \vdots \\ d_n(x, y) \end{bmatrix} \tag{38}$$

---

[3]One could alternatively consider using manifold optimization methods when $\mathcal{G}$ is a continuous manifold. However, these methods are somewhat cumbersome and aren't easy to extend to the case where $\mathcal{G}$ is a discrete set, such as the set of all permutations.

The triangle inequality now follows from:

$$d(x, y) = \sqrt[p]{d_1(x, y)^p + \ldots + d_n(x, y)^p} = \|\boldsymbol{d}(x, y)\|_p \tag{39}$$

$$\leq \|\boldsymbol{d}(x, m) + \boldsymbol{d}(m, y)\|_p \tag{40}$$

$$\leq \|\boldsymbol{d}(x, m)\|_p + \|\boldsymbol{d}(m, y)\|_p \tag{41}$$

$$= d(x, m) + d(m, y) \tag{42}$$

for all $x \in \mathcal{S}$, $y \in \mathcal{S}$, and $m \in \mathcal{S}$. The first inequality follows from the triangle inequality on each $d_1, \ldots, d_n$ and then from $\|\cdot\|_p$ being a monotonically increasing function of each vector coordinate. The second inequality follows from the sub-additivity property of all norms. $\square$

This lemma provides further perspective on the closed form expression we gave in appendix D.1 for the 2-Wasserstein distance between Gaussian distributions. In particular, if $P_i = \mathcal{N}(\boldsymbol{\mu}_i, \boldsymbol{\Sigma}_i)$ and $P_j = \mathcal{N}(\boldsymbol{\mu}_j, \boldsymbol{\Sigma}_j)$, we saw in equation (27) that:

$$\mathcal{W}_2\big(P_i, P_j\big) = \big(d_{\boldsymbol{\mu}}^2(\boldsymbol{\mu}_i, \boldsymbol{\mu}_j) + d_{\boldsymbol{\Sigma}}^2(\boldsymbol{\Sigma}_i, \boldsymbol{\Sigma}_j)\big)^{1/2} \tag{43}$$

where $d_{\boldsymbol{\mu}}^2$ is the squared Euclidean distance between the means and $d_{\boldsymbol{\Sigma}}^2$ is the squared Bures metric between the covariances. Thus, the 2-Wasserstein distance between Gaussians can be intuitively thought of as the $\ell_2$ norm of this pair of metrics.

It is trivial to verify that all the properties of a metric are preserved the distance is multiplied a scalar $\alpha > 0$. That is, if $g$ is a metric, then $d(x, y) = \alpha g(x, y)$ is also a metric. Combining this with lemma 3 it is obvious that,

$$\overline{\mathcal{W}}_2^{\alpha}\big(P_i, P_j\big) = \big(\alpha \cdot d_{\boldsymbol{\mu}}^2(\boldsymbol{\mu}_i, \boldsymbol{\mu}_j) + (2 - \alpha) \cdot d_{\boldsymbol{\Sigma}}^2(\boldsymbol{\Sigma}_i, \boldsymbol{\Sigma}_j)\big)^{1/2} \tag{44}$$

which is simply a re-statement of equation (7), is a metric for any $0 < \alpha < 2$.

### E.2 Lower Bound on Interpolated Shape Distances

We now derive a simple lower bound on stochastic shape distances when $\overline{\mathcal{W}}_2^{\alpha}$ is used as the ground metric. Let $d_{\alpha}^2$ denote the squared shape distance of interest, for any chosen $0 \leq \alpha \leq 2$. We have:

$$d_{\alpha}^2(F_i, F_j) = \min_{\boldsymbol{T} \in \mathcal{G}} \mathbb{E}_{\boldsymbol{z}}\left[(\overline{\mathcal{W}}_2^{\alpha})^2\big(F_i^{\phi}(\cdot \mid \boldsymbol{z}), F_j^{\phi}(\cdot \mid \boldsymbol{z}) \circ \boldsymbol{T}^{-1}\big)\right]$$

$$= \min_{\boldsymbol{T} \in \mathcal{G}} \mathbb{E}_{\boldsymbol{z}}\left[\alpha \cdot d_{\boldsymbol{\mu}}^2(\boldsymbol{\mu}_i(\boldsymbol{z}), \boldsymbol{T}\boldsymbol{\mu}_j(\boldsymbol{z})) + (2 - \alpha) \cdot d_{\boldsymbol{\Sigma}}^2(\boldsymbol{\Sigma}_i(\boldsymbol{z}), \boldsymbol{T}\boldsymbol{\Sigma}_j(\boldsymbol{z})\boldsymbol{T}^{\top})\right]$$

$$\geq \alpha \cdot \min_{\boldsymbol{T}_{\boldsymbol{\mu}} \in \mathcal{G}} \mathbb{E}_{\boldsymbol{z}}\left[d_{\boldsymbol{\mu}}^2(\boldsymbol{\mu}_i(\boldsymbol{z}), \boldsymbol{T}_{\boldsymbol{\mu}}\boldsymbol{\mu}_j(\boldsymbol{z}))\right] + (2 - \alpha) \cdot \min_{\boldsymbol{T}_{\boldsymbol{\Sigma}} \in \mathcal{G}} \mathbb{E}_{\boldsymbol{z}}\left[d_{\boldsymbol{\Sigma}}^2(\boldsymbol{\Sigma}_i(\boldsymbol{z}), \boldsymbol{T}_{\boldsymbol{\Sigma}}\boldsymbol{\Sigma}_j(\boldsymbol{z})\boldsymbol{T}_{\boldsymbol{\Sigma}}^{\top})\right]$$

The inequality here follows from the linearity of expectation and then from separately minimizing the two terms. The inequality is tight if the optimal value of $\boldsymbol{T}_{\boldsymbol{\mu}}$ equals the optimal value of $\boldsymbol{T}_{\boldsymbol{\Sigma}}$. Furthermore, the two minimized terms in the final expression are proportional to the squared shape distance when $\alpha = 2$ and $\alpha = 0$, respectively:

$$\min_{\boldsymbol{T} \in \mathcal{G}} \mathbb{E}_{\boldsymbol{z}}\left[d_{\boldsymbol{\mu}}^2(\boldsymbol{\mu}_i(\boldsymbol{z}), \boldsymbol{T}\boldsymbol{\mu}_j(\boldsymbol{z}))\right] = \frac{1}{2} \cdot d_{\alpha=2}^2(F_i, F_j) \tag{45}$$

$$\min_{\boldsymbol{T} \in \mathcal{G}} \mathbb{E}_{\boldsymbol{z}}\left[d_{\boldsymbol{\Sigma}}^2(\boldsymbol{\Sigma}_i(\boldsymbol{z}), \boldsymbol{T}_{\boldsymbol{\Sigma}}\boldsymbol{\Sigma}_j(\boldsymbol{z})\boldsymbol{T}_{\boldsymbol{\Sigma}}^{\top})\right] = \frac{1}{2} \cdot d_{\alpha=0}^2(F_i, F_j) \tag{46}$$

Thus, in summary we have:

$$d_{\alpha}^2(F_i, F_j) \geq \frac{\alpha}{2} \cdot d_{\alpha=2}^2(F_i, F_j) + \frac{2 - \alpha}{2} \cdot d_{\alpha=0}^2(F_i, F_j) \tag{47}$$

$$\Rightarrow d_{\alpha}(F_i, F_j) \geq \sqrt{\frac{\alpha}{2} \cdot d_{\alpha=2}^2(F_i, F_j) + \frac{2 - \alpha}{2} \cdot d_{\alpha=0}^2(F_i, F_j)} \tag{48}$$

## E.3 Interpretation of $\overline{\mathcal{W}}_2^\alpha$ when Gaussian assumption is violated

When neural responses are Gaussian-distributed, then $\overline{\mathcal{W}}_2^\alpha$ can be interpreted as a natural extension of 2-Wasserstein distance (see sec. 2.4). What if neural responses are *not* Gaussian-distributed? Concretely, consider two distributions $P_i$ and $P_j$, which are not necessarily Gaussian. We can still define the first two moments (mean and covariance) of these distributions:

$$\boldsymbol{\mu}_i = \mathbb{E}_{\boldsymbol{x} \sim P_i}[\boldsymbol{x}] \quad \text{and} \quad \boldsymbol{\Sigma}_i = \mathbb{E}_{\boldsymbol{x} \sim P_i}[(\boldsymbol{x} - \boldsymbol{\mu}_i)(\boldsymbol{x} - \boldsymbol{\mu}_i)^\top]. \tag{49}$$

Using these, we can still compute $\overline{\mathcal{W}}_2^\alpha(P_i, P_j)$ as before.

However, it is no longer the case that this calculation will coincide with the 2-Wasserstein distance between $P_i$ and $P_j$. That is, since the Gaussian assumption is unreliable, $\overline{\mathcal{W}}_2^{\alpha=1}(P_i, P_j) \neq \mathcal{W}_2(P_i, P_j)$. Because of this, we can no longer conceptualize $\overline{\mathcal{W}}_2^\alpha$ as the amount of energy taken to transport $P_i$ onto $P_j$ with the parameter $\alpha$ differentially weighting the cost of transporting mass due to mismatches in the mean and covariance.

On the other hand, $\overline{\mathcal{W}}_2^\alpha$ may still be a reasonable ground metric in many practical circumstances. In particular, it is obvious that $\overline{\mathcal{W}}_2^\alpha(P_i, P_j) = 0$ if and only if the mean and covariance of these distributions match. Thus, it is a pseudometric over all probability distributions and a metric on equivalence classes defined by the equivalence relation $P_i \sim P_j$ if and only if $\boldsymbol{\mu}_i = \boldsymbol{\mu}_j$ and $\boldsymbol{\Sigma}_i = \boldsymbol{\Sigma}_j$. From this, it is easy to show that the stochastic shape metric (eq. 5) based on this ground metric also satisfies the metric space axioms, including the triangle inequality.

In high-dimensional datasets, it is often challenging to estimate and interpret the higher-order statistical moments of a distribution. In the setting of comparing stochastic neural representations, one could argue that it is reasonable to settle for a ground metric that is insensitive to these higher-order moments. From this perspective, $\overline{\mathcal{W}}_2^\alpha$ belongs to a larger family of ground metrics that can be expressed:

$$\mathcal{D}(P_i, P_j) = (d_{\boldsymbol{\mu}}^2(\boldsymbol{\mu}_i, \boldsymbol{\mu}_j) + d_{\boldsymbol{\Sigma}}^2(\boldsymbol{\Sigma}_i, \boldsymbol{\Sigma}_j))^{1/2} \tag{50}$$

for some chosen metric on the means, $d_{\boldsymbol{\mu}}$, and another chosen metric on the covariances, $d_{\boldsymbol{\Sigma}}$. Again, no assumption on whether $P_i$ and $P_j$ being Gaussian is strictly necessary. A more thorough exploration of these alternative ground metrics is a potential direction of future research.

## F Miscellaneous Theory and Background

### F.1 Energy distance as a trial-averaged shape metric when $q = 2$

Performing representational dissimilarity analysis on trial-average activity measurements is already common practice in neuroscience. Here, we show that this approach arises as a special case of the stochastic shape distances explored in this manuscript. When $q = 2$, the energy distance is given by:

$$\mathcal{E}_2(P, Q) = (\mathbb{E}\|X - Y\|^2 - \tfrac{1}{2}\mathbb{E}\|X - X'\|^2 - \tfrac{1}{2}\mathbb{E}\|Y - Y'\|^2)^{1/2} \tag{51}$$

Since $X$ and $X'$ are independent and identically distributed random variables, we have:

$$\tfrac{1}{2}\mathbb{E}\|X - X'\|^2 = \tfrac{1}{2}\mathbb{E}[X^\top X] + \tfrac{1}{2}\mathbb{E}[X'^\top X'] - \mathbb{E}[X^\top X'] \tag{52}$$

$$= \mathbb{E}[X^\top X] - \mathbb{E}[X^\top X'] \tag{53}$$

$$= \mathbb{E}[X^\top X] - \mathbb{E}[X]^\top \mathbb{E}[X] \tag{54}$$

Likewise,

$$\tfrac{1}{2}\mathbb{E}\|Y - Y'\|^2 = \mathbb{E}[Y^\top Y] - \mathbb{E}[Y]^\top \mathbb{E}[Y]. \tag{55}$$

Plugging these expressions into equation (51) and simplifying we see that:

$$\mathcal{E}_2(P, Q) = (\mathbb{E}\|X - Y\|^2 - \mathbb{E}[X^\top X] + \mathbb{E}[X]^\top \mathbb{E}[X] - \mathbb{E}[Y^\top Y] + \mathbb{E}[Y]^\top \mathbb{E}[Y])^{1/2}$$

$$= (\cancel{\mathbb{E}[X^\top X]} + \cancel{\mathbb{E}[Y^\top Y]} - 2\mathbb{E}[X^\top Y] - \cancel{\mathbb{E}[X^\top X]} + \mathbb{E}[X]^\top \mathbb{E}[X] - \cancel{\mathbb{E}[Y^\top Y]} + \mathbb{E}[Y]^\top \mathbb{E}[Y])^{1/2}$$

$$= (\mathbb{E}[X]^\top \mathbb{E}[X] + \mathbb{E}[Y]^\top \mathbb{E}[Y] - 2\mathbb{E}[X^\top Y])^{1/2}$$

$$= (\|\mathbb{E}[X] - \mathbb{E}[Y]\|^2)^{1/2}$$

$$= \|\mathbb{E}[X] - \mathbb{E}[Y]\|$$

To summarize, we have shown that the $q = 2$ energy distance between a distribution $P$ and $Q$ is equal to the Euclidean distance between the mean of $P$ and the mean of $Q$. If we use this energy distance as the ground metric, $\mathcal{D}$, in proposition 1 to construct a stochastic shape distance, we are essentially calculating a deterministic shape distance[4] on the mean response patterns.

### F.2 ITERATIVELY REWEIGHTED LEAST SQUARES

Fix $q$ to be a value on the open interval $(0, 2)$ and consider the following optimization problem:

$$\min_{\boldsymbol{T} \in \mathcal{G}} \left\{ f(\boldsymbol{T}) = \sum_{i=1}^{N} \|\boldsymbol{y}_i - \boldsymbol{T}\boldsymbol{x}_i\|^q \right\} \tag{56}$$

It is easy to see that equation (36) is an instance of this problem for a particular choice of vectors $\boldsymbol{x}_i \in \mathbb{R}^n$ and $\boldsymbol{y}_i \in \mathbb{R}^n$.

Our key assumption will be that we can efficiently solve the following weighted least squares problem:

$$\min_{\boldsymbol{T} \in \mathcal{G}} \sum_{i=1}^{N} w_i \|\boldsymbol{y}_i - \boldsymbol{T}\boldsymbol{x}_i\|^2 \tag{57}$$

for any choice of weightings, $w_1, \ldots, w_N$. Again, this is possible when $\mathcal{G}$ is the orthogonal group (Procrustes problem) or the permutation group (linear assignment).

Iteratively re-weighted least squares algorithms are a family of methods that are encompassed by the even larger family of majorize-minimization algorithms (Lange, 2016). The specific method we deploy can be viewed as an extension to Weiszfeld's algorithm (Kuhn, 1973). Our starting point is to recognize that the function $s \mapsto s^{q/2}$ is concave for $0 < q < 2$ and $s \geq 0$. Thus, we can derive an upper bound using the first-order Taylor expansion:

$$(s + \delta)^{q/2} \leq s^{q/2} + \delta \left( \frac{\mathrm{d}}{\mathrm{d}s} s^{q/2} \right) = s^{q/2} + \frac{q}{2} \left( \frac{\delta}{s^{(1-q/2)}} \right), \tag{58}$$

for any $\delta$ such that $s + \delta \geq 0$.

We will now use this fact to derive an upper bound on the objective function in equation (56). Let $\boldsymbol{T}^{(t)} \in \mathcal{G}$ represent our estimate of the optimal $\boldsymbol{T} \in \mathcal{G}$ after $t$ iterations of our algorithm, and let $\boldsymbol{T} \in \mathcal{G}$ denote any feasible transformation. Then, for $i \in \{1, \ldots, N\}$, define:

$$s_i^{(t)} = \|\boldsymbol{y}_i - \boldsymbol{T}^{(t)}\boldsymbol{x}_i\|^2 \tag{59}$$

$$\delta_i^{(t)} = \|\boldsymbol{y}_i - \boldsymbol{T}\boldsymbol{x}_i\|^2 - s_i^{(t)} \tag{60}$$

Notice that these definitions imply $s_i^{(t)} + \delta_i^{(t)} \geq 0$. Now, plugging into equation (58), we have:

$$\left( s_i^{(t)} + \delta_i^{(t)} \right)^{q/2} = \|\boldsymbol{y}_i - \boldsymbol{T}\boldsymbol{x}_i\|^q \leq \left( s_i^{(t)} \right)^{q/2} + \frac{q}{2} \left( \frac{\delta_i^{(t)}}{\left( s_i^{(t)} \right)^{(1-q/2)}} \right) \tag{61}$$

This is an upper bound for each term in the sum of the original objective function. Therefore, plugging in the definitions of $s_i^{(t)}$ and $\delta_i^{(t)}$, we have:

$$f(\boldsymbol{T}) = \sum_{i=1}^{N} \|\boldsymbol{y}_i - \boldsymbol{T}\boldsymbol{x}_i\|^q \leq \sum_{i=1}^{N} \left( \|\boldsymbol{y}_i - \boldsymbol{T}^{(t)}\boldsymbol{x}_i\|^2 \right)^{q/2} + \frac{q}{2} \left( \frac{\|\boldsymbol{y}_i - \boldsymbol{T}\boldsymbol{x}_i\|^2 - s_i^{(t)}}{\left( \|\boldsymbol{y}_i - \boldsymbol{T}^{(t)}\boldsymbol{x}_i\|^2 \right)^{(1-q/2)}} \right) \tag{62}$$

$$= \sum_{i=1}^{N} \|\boldsymbol{y}_i - \boldsymbol{T}^{(t)}\boldsymbol{x}_i\|^q + \frac{q}{2} \left( \frac{\|\boldsymbol{y}_i - \boldsymbol{T}\boldsymbol{x}_i\|^2 - \|\boldsymbol{y}_i - \boldsymbol{T}^{(t)}\boldsymbol{x}_i\|^2}{\|\boldsymbol{y}_i - \boldsymbol{T}^{(t)}\boldsymbol{x}_i\|^{2-q}} \right) \tag{63}$$

$$\triangleq Q(\boldsymbol{T} \mid \boldsymbol{T}^{(t)}) \tag{64}$$

---

[4]Specifically, see the distances covered under Proposition 1 in Williams et al. (2021).

Here, we view $Q(\boldsymbol{T} \mid \boldsymbol{T}^{(t)})$ as a function of $\boldsymbol{T}$—i.e. $\boldsymbol{T}^{(t)}$ is fixed. The calculations above show that $Q(\boldsymbol{T} \mid \boldsymbol{T}^{(t)})$ provides an upper bound on the objective function for any $\boldsymbol{T} \in \mathcal{G}$. Furthermore, it is easy to check that $f(\boldsymbol{T}^{(t)}) = Q(\boldsymbol{T}^{(t)} \mid \boldsymbol{T}^{(t)})$—i.e., the upper bound is tight at $\boldsymbol{T} = \boldsymbol{T}^{(t)}$.

We now have all the necessary ingredients to derive an algorithm. We start by initializing $\boldsymbol{T}^{(1)} \in \mathcal{G}$ by some method. Then we compute $\{\boldsymbol{T}^{(2)}, \boldsymbol{T}^{(3)}, \ldots\}$ iteratively according to:

$$\boldsymbol{T}^{(t+1)} = \underset{\boldsymbol{T} \in \mathcal{G}}{\operatorname{argmin}} \, Q(\boldsymbol{T} \mid \boldsymbol{T}^{(t)}) = \underset{\boldsymbol{T} \in \mathcal{G}}{\operatorname{argmin}} \, \sum_{i=1}^{N} \frac{\|\boldsymbol{y}_i - \boldsymbol{T}\boldsymbol{x}_i\|^2}{\|\boldsymbol{y}_i - \boldsymbol{T}^{(t)}\boldsymbol{x}_i\|^{2-q}} \,. \tag{65}$$

The last equality here follows from dropping terms from equation (63) that are constant.[5] Intuitively, at each step we are minimizing a surrogate function $Q(\boldsymbol{T} \mid \boldsymbol{T}^{(t)})$ that upper bounds the true objective function. This surrogate function is easy to optimize since the minimization is a special case of equation (57) with weightings:

$$w_i = \frac{1}{\|\boldsymbol{y}_i - \boldsymbol{T}^{(t)}\boldsymbol{x}_i\|^{2-q}} \tag{66}$$

Furthermore, because we showed that the upper bound is tight at $\boldsymbol{T} = \boldsymbol{T}^{(t)}$, we have:

$$f(\boldsymbol{T}^{(t+1)}) \leq Q(\boldsymbol{T}^{(t+1)} \mid \boldsymbol{T}^{(t)}) = \min_{\boldsymbol{T} \in \mathcal{G}} Q(\boldsymbol{T} \mid \boldsymbol{T}^{(t)}) \leq Q(\boldsymbol{T}^{(t)} \mid \boldsymbol{T}^{(t)}) = f(\boldsymbol{T}^{(t)}) \tag{67}$$

which shows that the the objective function never increases as the algorithm progresses.

## F.3 QUADRATIC METRIC REPAIR

We are given a symmetric estimate of a distance matrix $\widetilde{\boldsymbol{D}} \in \mathbb{R}^{K \times K}$, which may contain negative entries and triangle inequality violations. Let $\widetilde{\boldsymbol{d}} \in \mathbb{R}^{K(K-1)/2}$ be a vector holding the upper triangular entries of $\widetilde{\boldsymbol{D}}$, excluding the diagonal. Then, consider the following optimization problem:

$$
\begin{aligned}
\underset{\boldsymbol{x}}{\text{minimize}} \quad & \|\boldsymbol{x} - \widetilde{\boldsymbol{d}}\|^2 \\
\text{subject to} \quad & x_i \geq 0, && \forall i \in \{1, \ldots, K(K-1)/2\} \\
& x_i + x_j - x_k \geq 0, && \forall (i,j,k) \in \mathcal{T}_K
\end{aligned}
$$

where $\mathcal{T}_K$ is the set of $3\binom{K}{3}$ directed triples of indices corresponding to a triangle inequality constraint. This is a quadratic program—i.e., a convex optimization problem with a quadratic objective and linear inequality constraints. To solve this problem, we use the open-source and highly optimized OSQP solver (Stellato et al., 2020). The number of inequality constraints grows cubically as $K$ increases, so finding an exact solution may be computationally expensive for analyses of large collections of stochastic neural networks.

## F.4 REFORMULATING THE BURES METRIC

Here we will prove that equations (28) and (29) are equivalent. A similar statement is proved in Theorem 1 of Bhatia et al. (2019). Our proof relies on the following lemma.

**Lemma 4.** *Let $\boldsymbol{X} \in \mathbb{R}^{n \times n}$ be a matrix with singular value decomposition $\boldsymbol{X} = \boldsymbol{U}\boldsymbol{S}\boldsymbol{V}^\top$. Then $\boldsymbol{V}\boldsymbol{U}^\top = (\boldsymbol{X}^\top\boldsymbol{X})^{-1/2}\boldsymbol{X}^\top$.*

*Proof.* This follows from the construction of the singular value decomposition. First, recognize that $\boldsymbol{X}$ can be written as the product of an orthogonal $\boldsymbol{Q}$ and symmetric positive semidefinite matrix, $\boldsymbol{P}$, as follows:

$$\boldsymbol{X} = \underbrace{\boldsymbol{X}(\boldsymbol{X}^\top\boldsymbol{X})^{-1/2}}_{=\boldsymbol{Q}}\underbrace{(\boldsymbol{X}^\top\boldsymbol{X})^{1/2}}_{=\boldsymbol{P}} \tag{68}$$

---

[5] It is important to understand that we are treating $\boldsymbol{T}^{(t)}$ as a constant. Only terms that depend on $\boldsymbol{T}$ matter for the minimization. We also further simplified by rescaling $Q(\boldsymbol{T} \mid \boldsymbol{T}^{(t)})$ by a factor of $2/q$, which doesn't affect the value at which the minimum is attained.

It is easy to check that $\boldsymbol{Q}^\top \boldsymbol{Q} = \boldsymbol{Q}\boldsymbol{Q}^\top = \boldsymbol{I}$. Now, since $\boldsymbol{P}$ is positive semidefinite, we have $\boldsymbol{P} = \boldsymbol{V}\boldsymbol{S}\boldsymbol{V}^\top$ for some orthogonal matrix $\boldsymbol{V}$ and nonnegative diagonal matrix $\boldsymbol{S}$. Defining $\boldsymbol{U} = \boldsymbol{Q}\boldsymbol{V}$, we arrive at the SVD of $\boldsymbol{X} = \boldsymbol{Q}\boldsymbol{P} = \boldsymbol{U}\boldsymbol{S}\boldsymbol{V}^\top$. Now we can see that:

$$\boldsymbol{U} = \boldsymbol{Q}\boldsymbol{V} = \boldsymbol{X}(\boldsymbol{X}^\top \boldsymbol{X})^{-1/2}\boldsymbol{V} \quad \Rightarrow \quad \boldsymbol{U}\boldsymbol{V}^\top = \boldsymbol{X}(\boldsymbol{X}^\top \boldsymbol{X})^{-1/2} \tag{69}$$

Taking the transpose of this we prove the lemma. $\qquad\square$

Now we proceed to prove the main result.

**Proposition 3.** *Let $\boldsymbol{A}$ and $\boldsymbol{B}$ be two positive definite matrices. Then*

$$\min_{\boldsymbol{Q} \in \mathcal{O}} \|\boldsymbol{A}^{1/2} - \boldsymbol{Q}\boldsymbol{B}^{1/2}\|_F^2 = Tr[\boldsymbol{A} + \boldsymbol{B} - 2(\boldsymbol{A}^{1/2}\boldsymbol{B}\boldsymbol{A}^{1/2})^{1/2}] \tag{70}$$

*Proof.* The minimization over $\boldsymbol{Q}$ is an instance of the well-known orthogonal procrustes problem (Gower & Dijksterhuis, 2004). This has a closed form solution. Specifically, denoting the singular value decomposition of $\boldsymbol{B}^{1/2}\boldsymbol{A}^{1/2}$ as $\boldsymbol{U}\boldsymbol{S}\boldsymbol{V}^\top$, we have:

$$\boldsymbol{Q}^* = \operatorname*{argmin}_{\boldsymbol{Q} \in \mathcal{O}} \|\boldsymbol{A}^{1/2} - \boldsymbol{Q}\boldsymbol{B}^{1/2}\|_F^2 = \boldsymbol{V}\boldsymbol{U}^\top \tag{71}$$

Now, by Lemma 4 above, we have:

$$\boldsymbol{V}\boldsymbol{U}^\top = ((\boldsymbol{B}^{1/2}\boldsymbol{A}^{1/2})^\top(\boldsymbol{B}^{1/2}\boldsymbol{A}^{1/2}))^{-1/2}(\boldsymbol{B}^{1/2}\boldsymbol{A}^{1/2})^\top = (\boldsymbol{A}^{1/2}\boldsymbol{B}\boldsymbol{A}^{1/2})^{-1/2}\boldsymbol{A}^{1/2}\boldsymbol{B}^{1/2} \tag{72}$$

Plugging this into the original problem, we have:

$$\|\boldsymbol{A}^{1/2} - \boldsymbol{Q}^*\boldsymbol{B}^{1/2}\|_F^2 = \mathrm{Tr}[\boldsymbol{A} + \boldsymbol{B} - 2\boldsymbol{A}^{1/2}\boldsymbol{Q}^*\boldsymbol{B}^{1/2}] \tag{73}$$

$$= \mathrm{Tr}[\boldsymbol{A} + \boldsymbol{B} - 2\boldsymbol{A}^{1/2}(\boldsymbol{A}^{1/2}\boldsymbol{B}\boldsymbol{A}^{1/2})^{-1/2}\boldsymbol{A}^{1/2}\boldsymbol{B}] \tag{74}$$

Due to the cyclic trace property, this becomes:

$$\mathrm{Tr}[\boldsymbol{A} + \boldsymbol{B} - 2(\boldsymbol{A}^{1/2}\boldsymbol{B}\boldsymbol{A}^{1/2})^{-1/2}\boldsymbol{A}^{1/2}\boldsymbol{B}\boldsymbol{A}^{1/2}] = \mathrm{Tr}[\boldsymbol{A} + \boldsymbol{B} - 2(\boldsymbol{A}^{1/2}\boldsymbol{B}\boldsymbol{A}^{1/2})^{1/2}] \tag{75}$$

as claimed. $\qquad\square$

