# OpenReview forum: "Representational Dissimilarity Metric Spaces for Stochastic Neural Networks"
_ICLR.cc/2023/Conference — ICLR 2023 poster_

### Official Review · Reviewer_pofg · 2022-10-24

**Confidence:** 4
**Correctness:** 3
**Technical Novelty And Significance:** 3
**Empirical Novelty And Significance:** 3
**Recommendation:** 6

**Clarity, Quality, Novelty And Reproducibility:**

The writing of the paper is generally clear.
The novelty is moderate.

**Strength And Weaknesses:**


Strength:
Nice combination of theory and experiments.
The numerical experiments are extensive.
The theoretical framework is elegant and will be useful in some applications.

A few concerns:

Although the numerical experiments covered several examples,  the insights revealed were quite limited. Some of the results could be obtained by using fairly standard methods to study the mean and the noise covariance structure (e.g., Section 3.2).

The Gaussian assumption needs to be better justified, perhaps through more simulation experiments.  Having a small number of trials in practice is not a good *justification* of the Gaussian assumption. The maximum entropy argument is also problematic as the maximum entropy distribution depends on the domain being studied. For variables that are non-negative, the maximum entropy distribution will not be Gaussian.

The innovation on the “stochastic representation” needs to be toned down. This is not a new concept. Every general model with noise can be naturally treated as a “stochastic representation”. The classic RSA approaches can also deal with stochastic presentations, under well-defined noise assumptions (e.g., reviewed recently in Kriegeskorte & Wei, Nature Reviews Neuroscience, 2021).

The following reference should be cited and discussed, as it is closely related to the current work:
Shahbazi, Mahdiyar, et al. "Using distance on the Riemannian manifold to compare representations in brain and in models." NeuroImage 239 (2021): 118271.


**Summary Of The Paper:**

The submission entitled “Representational dissimilarity metric spaces for stochastic neural networks” proposed a new method to characterize the representational similarity of neural networks, taking into account the noise characteristics. The paper considered metrics over the stochastic shape using p-Wasserstein distance and some variations. Overall, this is a well-written paper. I enjoyed reading it.

**Summary Of The Review:**

Solid paper. The study is well executed with moderate innovation.

---

> ### Author Response · Authors · 2022-11-17
> **Responses to Weaknesses**
>
> Thank you for your feedback on our manuscript and for your positive comments on the clarity of writing and the elegance of the theoretical framework. We believe were able to address all of your major concerns in our uploaded revision. Some specific responses are summarized below.
>
> ---
>
> > *Although the numerical experiments covered several examples, the insights revealed were quite limited. Some of the results could be obtained by using fairly standard methods to study the mean and the noise covariance structure (e.g., Section 3.2).*
>
> The reviews have made it clear that section 3.2 was poorly explained in our original submission (see also our response to the AC). We have entirely re-written this section and we have streamlined the results (removing two panels from the original figure) to focus on a set of three specific take-home messages. We have highlighted the novel aspects of our analysis—i.e., that we compare how similar noise covariance geometry is across animals, which we believe can’t be easily done with standard methods.
>
> Overall, we agree that section 3.2 only provides a short and preliminary analysis. Nonetheless, we believe it shows that the geometry of noise covariance in biological data has the potential to show interesting structure—for example, it is not entirely redundant with the mean response geometry as one might expect if each neuron has independent Poisson noise (mean = variance; no off-diagonal noise correlations). At the very least, this is a promising start to show that neuroscientists may take interest in our method.
>
> > *The Gaussian assumption needs to be better justified... Having a small number of trials in practice is not a good justification of the Gaussian assumption. The maximum entropy argument is also problematic... for variables that are non-negative, the maximum entropy distribution will not be Gaussian.*
>
> We have provided an extended discussion of this in our revision, where we have noted that the maximum entropy justification is commonly invoked by people in the field, but is not fully rigorous. Here are the main points to consider with respect to the Gaussian approximation:
> * We view stochastic shape metrics as a broadly applicable tool. We recognize that in certain cases the Gaussian assumption may not be justified. In those cases, we suggest using the energy distance ground metric, which does not rely on this assumption. Supplemental Figures 4 and 7 show results using this alternative method.
> * Even if the Gaussian assumption is not precisely satisfied, one may still be interested in comparing the top two statistical moments (mean and covariance) across different networks. In this case, equation (6) is still a pseudometric over probability distributions (equal to zero if and only if two distributions have the same mean and covariance). We unfortunately lose the ability to interpret equation (6) as the Wasserstein distance in this case (e.g. the earth movers interpretation fails). Nonetheless the shape distance is still a reasonable thing to compute. We have now made this technical point clear in the revision (see Appendix E.3, referenced on page 4 of the revision).
> * There are cases in the machine learning literature of interest where the Gaussian assumption is satisfied exactly. For example, the analysis of VAEs in Figure 4 of our paper.
> * In neuroscience, the Gaussian assumption will never hold exactly (e.g. due to nonnegativity constraints on firing rates). Nonetheless, it is a very common assumption in the literature that can be reasonable under certain circumstances (e.g. large firing rates). For example, this assumption is invoked frequently in Kriegeskorte & Wei’s Nature Reviews Neuroscience piece you mention below.
> * A future extension of our framework to the case of Poisson noise would be of great interest to neuroscientists. We believe this may be possible.

---

> > ### Author Response · Authors · 2022-11-17
> > **Response continued...**
> >
> > > *The innovation on the “stochastic representation” needs to be toned down. This is not a new concept. Every general model with noise can be naturally treated as a “stochastic representation”. The classic RSA approaches can also deal with stochastic presentations, under well-defined noise assumptions (e.g., reviewed recently in Kriegeskorte & Wei, Nature Reviews Neuroscience, 2021).*
> >
> > We did not mean to imply that we are the first to conceptualize neural responses as stochastic distributions. We agree this is extremely well established within the field. To acknowledge this, we had cited a different review in the introduction (Kriegeskorte & Douglas), but we agree that adding a citation to Kriegeskorte & Wei is warranted in this paragraph. We also cited a select number of well-known “classic” papers from machine learning (Sietsma & Dow, 1991) and neuroscience (Shadlen et al., 1996; Abbott & Dayan, 1999). We have carefully edited our revision to tone down language as much as possible—e.g. In the introduction, we changed “We can conceptualize stochastic neural networks as…” -> “Stochastic neural networks may be conceptualized as…”
> >
> > We want to be very clear about what our contribution is. Our contribution is to provide a method to quantify distances between stochastic representations without averaging away noise. To our knowledge, this is the first method to make this comparison across networks whose units are mismatched by permutation or rotation. Our understanding is that existing RSA approaches are noise-aware but do not quantify distances in noise correlation geometry or the geometry of higher-order statistical moments. In section 4 we reviewed several notable RSA papers that deal with noise—Cai et al. (2016), Walther et al. (2016), and Diedrichsen et al. (2020). All of these papers deal with different ways of normalizing data, or adding Bayesian priors to reduce the sensitivity of RSA to trial-to-trial noise. The notion of dissimilarity they aim to measure only depends on the mean neural response—unlike our approach, they do not capture geometric differences in the orientation of noise. **We have now shown this explicitly in Supplemental Figure 3**. We made an honest effort to cover the range of ideas presented in this literature, but if there are specific papers that we missed or misinterpreted we would very much appreciate the reviewer’s further feedback.
> >
> > > *The following reference should be cited and discussed, as it is closely related to the current work: Shahbazi, Mahdiyar, et al.*
> >
> > Thank you for bringing this to our attention. We should have cited this in the original submission. Our revision cites Shahbazi et al. (2021) prominently in the first paragraph of the introduction alongside the other deterministic dissimilarity measures—CKA, RSA, CCA, shape metrics.

---

> > > ### Comment · Reviewer_pofg · 2022-11-28
> > > **the revision further improves the manuscript**
> > >
> > > I appreciate the authors' response and their revision. The revised version is further improved. While the improvement was insufficient for me to bump up my score from 6 to 8, I'd more strongly vote for the acceptance of this submission.

---

### Official Review · Reviewer_Moyn · 2022-10-27

**Confidence:** 3
**Correctness:** 4
**Technical Novelty And Significance:** 4
**Empirical Novelty And Significance:** 2
**Recommendation:** 8

**Clarity, Quality, Novelty And Reproducibility:**

The extension of representational similarity analyses to covariance structure is novel and important in my opinion. I don’t think there are any particularly important or novel findings from their analyses applying this method.

I found the paper clear and easy to follow.

The paper generally seems of high quality, and I don’t have any particular concerns about reproducibility.

I did not have the appropriate expertise to check the appendix proofs.


**Strength And Weaknesses:**

Strengths

The paper addresses an interesting and important topic: how to deal with stochasticity when quantifying large-scale similarity measures between biological or artificial networks. As noted, virtually all prior work has focused on the deterministic case, with comparatively superficial treatments of noise covariance structure (e.g., mahalanobis-based correction metrics).

I found the writing clear, logical, and easy to follow.

Applying their method to both biological and artificial networks is a strength in my opinion.

The method seems like it scales well to large datasets and many networks though not much information is reported about computation time as far as I can tell.

Weaknesses

All the metrics are motivated by the desire to reproduce a sensible Euclidean metric in the deterministic limit. This seems like a potentially desirable property, but it doesn’t say anything about how metric behaves in the stochastic case, which is the whole point. In Figure 1, they illustrate how the covariance structure of a set of responses can impact discriminability, but it is not obvious how their metric relates to the stimulus discriminability. This strikes me as a major shortcoming, since it makes it hard to understand what the metric tells you.

Their Gaussian-approximated Wasserstein metric allows for a rotation of the covariance structure. This rotation appears to be separate from any rotation allowed to align the two networks. It is not clear to me why this is appropriate or desirable. Is this partly why the mean-insensitive version of their metric does not produce the desired result in their toy dataset?

The experiments are useful demonstrations of the method, but don’t seem to shed new light on brain or network function. The fact that stimulus-driven variation is greater for natural stimuli doesn’t seem particularly surprising and presumably explains the findings in Figure 3B/3C. It is also not clear that their analysis tells us much about noise structure. In Figure 3F, what is each datapoint? A session? Why would we expect distances to covary across sessions between natural scenes and gratings? Do the neurons differ across sessions?

One thing that would be interesting is to know is the extent to which the noise covariance structure of a visual region (e.g., V1) is stereotyped, in the sense of being similar across animals but distinct across regions, and whether this structure is predictable from the stimulus-driven component. This seems like the type of question their method might be able to answer.

The authors seem to make a big deal out of whether distances are above or below the diagonal when plotting the covariance- and mean-insensitive measures against each other (e.g., Fig 3B/C and 4B). It’s not clear to me that this has any particular significance. Why are squared distances between the means comparable to the Frobenius norm of the covariances?

I can’t tell if the fact that you can decode network properties from their measures is notable (e.g., Fig 4). There is no baseline comparison against any alternative model.


**Summary Of The Paper:**

This paper introduces proper distance metrics for quantifying the similarity of representations in stochastic networks, either biological or artificial. The paper builds on a recent article by Williams, which described metrics for quantifying similarity in deterministic networks by allowing for a set of preprocessing steps (e.g., whitening) and alignment operations (e.g., rotation) before applying a “ground metric” (e.g., Euclidean norm). This paper uses the same framework but applies it to a set of probabilistic ground metrics that were chosen to converge to the Euclidean norm in cases where the network is deterministic. The paper focuses on a Gaussian approximation to one of these metrics (Wassertein distance). The Gaussian approximation results in a metric that has two terms, one based on the similarity of the means, which might be thought of as the deterministic component, and one based on the similarity of the covariances after rotational alignment. They also describe a meta-metric that linearly interpolates between these terms in order to emphasize either the means or covariances.

The authors apply their framework to a set of toy networks that differ in their covariance scale and correlational structure and report that their metrics can uncover the latent structure (Fig 2C). They then apply their approach to neural recordings from mouse primary visual cortex to gratings and natural scenes (Fig 3). They find that their similarity metric is dominated by the noise covariance structure for static gratings but not for natural scenes, presumably because natural scenes drive more variation in the mean response. The authors then apply their method to a large collection of 1800 VAEs, and report that their distance metrics can be used to decode a variety of network properties with varying degrees of accuracy (Fig 4). Finally, the apply their model to a Patch-Gaussian network trained with varying amounts and sizes of Gaussian noise, and report that their metrics can recover this structure to some degree (Fig 5).


**Summary Of The Review:**

The paper develops a framework to extend representational similarity to the stochastic regime, which I feel is an important contribution. The metric is solely motivated by how it behaves in the deterministic case which is a notable limitation in my opinion. There are no major novel findings from their analyses applying their method, but the analyses do serve to illustrate and validate the approach. The paper generally seems of high quality.

Post-rebuttal comments:

I apologize for my slow response. I thank the authors for taking so much time to consider and respond to my questions and comments. I was impressed with all of their responses including to other reviewers. I have upgraded my score from a 6 to an 8. I hope to see this paper accepted.

---

> ### Comment · Area_Chair_G9k6 · 2022-11-02
> **Wasserstein distance between Gaussians**
>
> > Their Gaussian-approximated Wasserstein metric allows for a rotation of the covariance structure. This rotation appears to be separate from any rotation allowed to align the two networks. It is not clear to me why this is appropriate or desirable.
>
> I also do not understand how the expression (6) can be correct. The Wasserstein distance and the Bures metric $\mathcal{B}$ between positive definite matrices (28) discussed in the appendix are supposed to be metrics. The expression (6) vanishes for Gaussians differing by a rotation, i.e. having same means but different covariance matrices $\Sigma_1$ and  $\Sigma_2 = U \Sigma_1 U^T \neq \Sigma_1$ for some rotation $U$. So (6) is not a metric. Where is the catch?

---

> > ### Author Response · Authors · 2022-11-07
> > **Equation 6 is correct, and is a metric**
> >
> > Thank you for taking a close look at our submission. We are currently writing a comprehensive response and revision to the feedback provided by the reviewers and area chair. This feedback is valuable and addressable within the 2 week timeframe, and we believe it will lead to a much stronger final paper.
> >
> > To avoid any misunderstanding, we wanted to post as soon as possible to confirm that Equation 6 is correct. As cited in our paper, please see Theorem 1 on Page 2 of [Bhatia et al. (2019)](https://arxiv.org/abs/1712.01504), which proves that the square of the Bures metric can be written:
> > \begin{equation}
> > \mathcal{B}^2(A, B) = \min_{U \in \mathcal{O}} \Vert A^{1/2}  - U B^{1/2} \Vert_F^2 = \textrm{Tr}[A + B - 2 (A^{1/2} B A^{1/2} )^{1/2} ]
> > \end{equation}
> > We have also provided a self-contained proof at the bottom of this post, which we will append to the supplemental materials of our final submission.
> >
> > We stress that the minimization over orthogonal matrices $U \in \mathcal{O}$ arises for entirely different reasons than the minimization over "nuisance transformations" (which is $T \in \mathcal{G}$ in the notation of our paper). We thank the reviewer for bringing it to our attention that this technical detail is tricky to understand. In our revision, we will make sure to edit this portion of the text to emphasize that $U$ should not be interpreted as an alignment over nuisance transformations and plays a different role than $T$.
> >
> > We also want to clear up the area chair's comment "The expression (6) vanishes for Gaussians differing by a rotation." This is not true because the rotation $U$ is only applied to one side of the square root covariance. First, note that the pushforward measure of a normal distribution by a rotation $Q$ is given by $\mathcal{N}(\mu, \Sigma) \circ Q^{-1} = \mathcal{N}(Q \mu, Q \Sigma Q^\top)$. Then, referring to equation (6), the minimized expression is $\Vert A^{1/2}  - U B^{1/2} \Vert_F^2$. **We are not minimizing** $\Vert A^{1/2}  - U B^{1/2} U^\top \Vert_F^2$. If the latter expression were to be minimized then the area chair would be correct that equation (6) is be zero between $\mathcal{N}(0, Q \Sigma Q^\top)$ and $\mathcal{N}(0, \Sigma)$ for any orthogonal matrix $Q$.
> >
> > Again, we very much understand this detail is tricky and subtle and we appreciate you bringing it to our attention so that we can improve the clarity of the manuscript.
> >
> > ------
> >
> > Here is a quick proof of the claim above (see also Theorem 1 in Bhatia et al.). We start by proving the following lemma.
> >
> > **Lemma:** Let $X \in \mathbb{R}^{n \times n}$ be a full rank matrix with singular value decomposition $X = U S V^\top$. Then $V U^\top = (X^\top X)^{-1/2} X^\top$.
> >
> > **Proof of Lemma:** The SVD is derived by noticing that $X$ can be written as the product of an orthogonal $Q$ and symmetric positive definite matrix, $P$, as follows:
> > \begin{equation}
> > X = \underbrace{X (X^\top X)^{-1/2}}_{Q} \underbrace{(X^\top X)^{1/2}}_P
> > \end{equation}
> > It is easy to check that $Q^\top Q = Q Q^\top = I$. Now, since $P$ is positive definite, we have $P = V S V^\top$ for some orthogonal matrix $V$. Defining $U = Q V$, we arrive at the SVD of $X = Q P = U S V^\top$.
> > Now we can see that:
> > \begin{equation}
> > U = Q V = X (X^\top X)^{-1/2} V \quad \Rightarrow \quad U V^\top = X (X^\top X)^{-1/2}
> > \end{equation}
> > Taking the transpose of this we prove the lemma.
> >
> > **Theorem:** Let $A$ and $B$ be two positive definite matrices. Then
> > \begin{equation}
> > \min_{Q \in \mathcal{O}} \Vert A^{1/2}  - Q B^{1/2} \Vert_F^2 = \textrm{Tr}[A + B - 2 (A^{1/2} B A^{1/2} )^{1/2} ]
> > \end{equation}
> >
> > **Proof:** Applying SVD, $B^{1/2}A^{1/2} = U S V^\top$. Then it is well-known that the optimal value of $Q$, which we denote $Q^*$, is given by $V U^\top$. (For proof, see [wiki page on orthogonal procrustes](https://en.wikipedia.org/wiki/Orthogonal_Procrustes_problem).) By the Lemma above, we have:
> > \begin{equation}
> > Q^* = V U^\top = ((B^{1/2} A^{1/2})^\top (B^{1/2} A^{1/2}))^{-1/2} (B^{1/2} A^{1/2})^\top = (A^{1/2} B A^{1/2})^{-1/2} A^{1/2} B^{1/2}
> > \end{equation}
> > Plugging this in, we have:
> > \begin{equation}
> > \Vert A^{1/2}  - Q^* B^{1/2} \Vert_F^2 = \textrm{Tr}[ A + B - 2 A^{1/2} Q^* B^{1/2} ] = \textrm{Tr}[ A + B - 2 A^{1/2} (A^{1/2} B A^{1/2})^{-1/2} A^{1/2} B]
> > \end{equation}
> > Due to the cyclic trace property, this becomes:
> > \begin{equation}
> > \textrm{Tr}[ A + B - 2 (A^{1/2} B A^{1/2})^{-1/2} A^{1/2} B A^{1/2} ] = \textrm{Tr}[ A + B - 2 (A^{1/2} B A^{1/2})^{1/2} ]
> > \end{equation}
> > as claimed.

---

> > > ### Comment · Area_Chair_G9k6 · 2022-11-08
> > > **Clarified**
> > >
> > > I thank the authors for clarification and apologize for my hasty initial consideration of this identity.

---

> ### Author Response · Authors · 2022-11-17
> **Responses to Weaknesses**
>
> We thank the reviewer for their detailed comments and feedback. We appreciate that you see our theoretical extension of representational similarity to the stochastic regime as an important contribution. We agree this is the core contribution and that our experiments are only meant to serve as preliminary appetizers for deeper investigations to come. We believe we have addressed all of your concrete concerns, as detailed below.
>
> ---
>
> > *The method seems like it scales well to large datasets and many networks though not much information is reported about computation time as far as I can tell.*
>
> We have amended our Appendix with a discussion of implementation and algorithmic complexity in subsection D1.1. and refer to it in section 2.3 of the main text.
>
> > *All the metrics are motivated by the desire to reproduce a sensible Euclidean metric in the deterministic limit. This seems like a potentially desirable property, but it doesn’t say anything about how metric behaves in the stochastic case.*
>
> It is true that this motivation alone does not provide intuition into how the metric behaves in the stochastic case. To build some of this intuition we provided an analysis of a toy dataset (Figure 2). We agree that a more pedagogical exposition would be valuable and have done our best to accomodate this in the revision.
>
> Supplemental Figure 1 in our revision helps to unpack the definitions of Wasserstein and Energy distance (equations 3 and 4). These are well-studied distances and are actually very intuitive once the right background is developed. For example, the [Wasserstein distance](https://en.wikipedia.org/wiki/Wasserstein_metric) can be conceptualized as the amount of energy it takes to move one pile of dirt (the density of P) into a different configuration (the density of Q) when one uses an optimal transport plan. The energy distance can be conceptualized as the amount of energy it takes when one uses a maximally random (i.e. maximum entropy) transport plan. See [Feydy et al. (2018)](https://arxiv.org/abs/1810.08278) for more details.
>
> Supplemental Figure 2 also provides some useful intuition in a different toy dataset.
>
> > *In Figure 1, they illustrate how the covariance structure of a set of responses can impact discriminability, but it is not obvious how their metric relates to the stimulus discriminability.*
>
> We agree it would be desirable to relate shape distances to other quantities like discriminability. Stochastic shape metrics provide a global measure of dissimilarity between two representations (i.e. whether the representation of K stimulus classes are arranged similarly between two networks), whereas discriminability is a local measure (e.g. how perturbations to a single stimulus impact its representation). Both are useful and are studied by complementary papers in the literature.
>
> Currently we don’t know how to rigorously relate representational distance to discriminability either for stochastic metrics or for existing methods in the literature (deterministic shape metrics, RSA, CKA, etc.). However, our framework seems more promising because it can at least distinguish between stochastic representations with different degrees of discriminability (see the toy data in Figure 2 vs. the comparisons in Supp. Fig. 3). Future work relating global structure (e.g. representational dissimilarity) to local structure (e.g. discriminability) is an interesting direction for future research.
>
> > *Their Gaussian-approximated Wasserstein metric allows for a rotation of the covariance structure. This rotation appears to be separate from any rotation allowed to align the two networks. It is not clear to me why this is appropriate or desirable. Is this partly why the mean-insensitive version of their metric does not produce the desired result in their toy dataset?*
>
> This is a misunderstanding. Please see our long response here to the follow-up question provided by the AC: https://openreview.net/forum?id=xjb563TH-GH&noteId=TAkjb007pyq
>
> To summarize, the same alignment, $\mathbf{T}$, is indeed applied to the full distribution (mean + covariance) when aligning across networks. There is a second rotation, $\mathbf{U}$, that appears in the definition of the Wasserstein distance (see eq. 6), but this is entirely unrelated to the alignment over nuisance transformations. We have included the proof linked above in Appendix F.6 of our revised paper.
>
> We thank the reviewer for bringing it to our attention that this technical detail is tricky to understand. We have edited this portion of the text to emphasize that $\mathbf{U}$ should not be interpreted as an alignment over nuisance transformations and plays a different role than $\mathbf{T}$. Furthermore, we’ve added further citations and added Supplemental Figure 1 which schematizes the full rotation being applied to both mean and covariance (see also Fig. 1E).

---

> > ### Author Response · Authors · 2022-11-17
> > **Responses Continued....**
> >
> > > *The experiments are useful demonstrations of the method, but don’t seem to shed new light on brain or network function.*
> >
> > We are a bit more optimistic about our results, but we largely we agree. It is not the main point of our paper to provide deep insight into new brain/network functions. The main point of our paper is theoretical and methodological. We do feel we've described an important set of analysis methods that have the capability to reveal important future insights into brain/network functions.
> >
> > > *The fact that stimulus-driven variation is greater for natural stimuli doesn’t seem particularly surprising and presumably explains the findings in Figure 3B/3C.*
> >
> > As mentioned elsewhere, we have substantial improved the writing and clarity of this experimental section in the revision. We aren’t exactly sure what the reviewer means by "stimulus-driven variation" and how this is an explanation for our findings. What our findings show is that mean response geometry (i.e. deterministic representation) tends to be more variable across animals than covariance geometry (mean-subtracting each stimulus) in naturalistic images. In artificial grating stimuli, the mean geometry is more similar across animals, but the covariance geometry is still variable.
> >
> > We think what the reviewer is getting at is that the trial-average neural responses are more “rich” and high-dimensional in naturalistic stimuli, so the variability across animals may be expected to go up as this richer stimulus set reveals more idiosyncrasy in each recording. But we don’t think it is obvious why similar idiosyncrasies would not be revealed within the covariance geometry. That is, it would have been possible (or perhaps even most expected) to see both the mean and covariance shape distances increase moving from gratings to natural scenes, but to have their ratio stay constant.
> >
> > Overall, we think that the fact this ratio isn't constant is an interesting observation that merits future exploration. We are not sure how one would observe this without the framework we developed in our paper. We plan to carry out a future study on this and potentially submit our findings to a more specialized, domain-specific neuroscience journal.
> >
> > > *In Figure 3F, what is each datapoint? A session?*
> >
> > We have removed this panel in the revision to streamline this section of the paper. We now focus on a more coherent set of take-home messages which are clearly enumerated. We encourage the reviewer to briefly re-read the revised section which we think is much more clear.
> >
> > Nonetheless, we want to answer directly. Each dot is a pairwise distance between two recording sessions (i.e. an element of the upper-triangle of the distance matrix across animals). Our intention was to ask the question -- if two animals have similar representations (either in terms of mean or covariance geometry) in one set of sensory stimuli (e.g. gratings), does this mean they will have similar representations on a different set of stimuli (e.g. natural images). This relates to outstanding debates in the field (Felsen and Yang, 2005; Rust and Movshon, 2005).
> >
> > > *Do the neurons differ across sessions?*
> >
> > Yes, we have clarified that each recording session is performed in a different animal.
> >
> > > *One thing that would be interesting is to know is the extent to which the noise covariance structure of a visual region (e.g., V1) is stereotyped, in the sense of being similar across animals but distinct across regions, and whether this structure is predictable from the stimulus-driven component. This seems like the type of question their method might be able to answer.*
> >
> > We agree this is a very interesting question and thank the reviewer for highlighting an example showing how our theoretical contributions will be practical. We have already begun an investigation into this question in the Allen Brain Observatory dataset.
> >
> > Ultimately, we decided this is outside the scope of the current study. The feedback from the AC + reviewers suggests that we tried to pack in too many example applications to the paper.

---

> > > ### Author Response · Authors · 2022-11-17
> > > **Responses Continued...**
> > >
> > > > *The authors seem to make a big deal out of whether distances are above or below the diagonal when plotting the covariance- and mean-insensitive measures against each other (e.g., Fig 3B/C and 4B). It’s not clear to me that this has any particular significance. Why are squared distances between the means comparable to the Frobenius norm of the covariances?*
> > >
> > > This is a great question. First, we want to emphasize that the covariance distances are the Bures metric (result of the min over $\mathbf{U}$ in Eq. 7; see Appendix D.1 for more detail and F.4 for self-contained proof). It is *not* simply the Frobenius norm of the difference in covariance matrices.
> > >
> > > To understand why these two terms are comparable, we first need to understand how to [interpret the Wasserstein distance in terms of optimal transport](https://en.wikipedia.org/wiki/Wasserstein_metric#Intuition_and_connection_to_optimal_transport). We have provided this intuition in Supplementary Figure 1 and by providing additional explanation in the main text. As explained above, we can think of Wasserstein distance as the amount of energy it takes to move two piles of dirt / probability distributions onto each other.
> > >
> > > Thus, the Bures metric can be interpreted as how much energy it takes to rotate and re-scale two Gaussian distributions that are centered at the origin. This is because the overall Wasserstein distance for Gaussians decomposes into a term that depends on the means (Euclidean distance) and a term that depends on the covariances (Bures distance). Comparing these two terms is therefore quite intuitive because it allows us to evaluate how much of the cost of transporting dirt is due to the covariances being different vs. the means being different.
> > >
> > > Additionally, we tried not compare these two distances in a vacuum, but rather investigate how their ratio changes across different stimulus conditions (Fig 3C-E), and pre vs post training (Fig 4B). Thus, it is not necessarily important that points fall above or below the diagonal per se, but that these distances appear to be sensitive to some experimental manipulation. Overall, we think this demonstrates how future studies might use stochastic shape metrics to quantify systematic differences in neural representations across experimental conditions.
> > >
> > > > *I can’t tell if the fact that you can decode network properties from their measures is notable (e.g., Fig 4). There is no baseline comparison against any alternative model.*
> > >
> > > To our knowledge, the only example within the published literature for decoding network properties from representational dissimilarity scores is done in the deterministic shape metrics paper (Williams et al. 2021). This baseline is included as a special case of our framework (namely, when alpha = 2). Importantly, the K-nearest neighbors approach we use to perform these regression requires the triangle inequality, and this hasn’t been emphasized until recent work by Williams et al. + Shahbazi et al.
> > >
> > > On a qualitative level, Locatello et al. did notice that different random seeds resulted in different factor disentanglement scores. We believe the findings using methods developed in our paper complement their analyses, and help make their observations more precise.

---

### Official Review · Reviewer_yN72 · 2022-11-04

**Confidence:** 3
**Clarity, Quality, Novelty And Reproducibility:** Clarity, quality, novelty and reprodu…
**Correctness:** 4
**Technical Novelty And Significance:** 3
**Empirical Novelty And Significance:** 3
**Recommendation:** 8

**Details Of Ethics Concerns:**

None.

**Strength And Weaknesses:**

Strengths:

- The problem of measuring distances between representations in stochastic networks is an important one, both in neuroscience and machine learning.  The formal properties investigated by the authors are well motivated, such as invariance to network permutations.  The authors offer a clean and general solution to this problem.
- Some interesting observations are made in the experimentation, which I'm not aware of having being observed before.  Particularly, the observation that the response of biological networks (using Allen mouse brain observatory data) to different classes of responses can generate representations primarily focused on the mean or the covariance (noise) structure, and that this is mirrored in artificial networks pre and post training.  Also, the observation that many artificial network training properties, such as random seed, reconstruction loss and regularization can be predicted from a network's noise structure is unexpected, and may be relevant for instance in predicting which networks will generalize well (e.g. in statistical learning theory).

Weaknesses:

- It is unclear whether the 'minimization' operation over the group of nuisance transformations is necessarily the right way of assessing distances between stochastic representations (which is used since the work generalizes previous work on deterministic shape metrics).  Potentially, some form of uncertainty about the best alignment should also be preserved over the group of nuisance transformations.
- The empirical work is somewhat preliminary, in the sense that many of the observations could be followed up in more detail with more substantial studies.  It is also limited in the sense that only covariance (pairwise) structure is considered when comparing 'noise' distributions empirically, while the framework would allow for higher-order properties of the distributions to be compared.

**Summary Of The Paper:**

The paper presents a novel class of stochastic shape metrics, which are suitable for assessing similarities in the representations of both biological and artificial stochastic networks.  The paper presents a thorough analysis of the formal properties of their class of metrics, showing they satisfy the metric axioms, that they have other desirable formal properties, and their relationship to certain previous metrics.  In their experimentation, they focus on a generalized form of 2-Wasserstein distance, which allows a fine-tuning of its sensitivity to mean vs covariance structure, and has a tractable analytic estimator.  They provide a series of case studies demonstrating the use of this metric on synthetic, biological and artificial networks, leading to some intriguing observations.

**Summary Of The Review:**

I recommend acceptance of the paper, since it offers a general solution to an important problem that will be of interest both to the neuroscience and machine learning communities.  Further, the results will be of interest to both communities (although, as noted above, several of the observations could perhaps be followed up in more detail in making the case for the use of generalized forms of stochastic metrics).

---

> ### Author Response · Authors · 2022-11-17
> **Responses to Weaknesses**
>
> We thank the reviewer for their constructive feedback and positive comments on the manuscript. We have only brief comments on the listed weaknesses.
>
> > *It is unclear whether the 'minimization' operation over the group of nuisance transformations is necessarily the right way... Potentially, some form of uncertainty about the best alignment should also be preserved...*
>
> This is a very interesting suggestion and is a current direction of research for us. There are both theoretical and practical challenges that we still need to overcome. The minimization is currently required to ensure the triangle inequality is satisfied (see Lemma 1 in the Supplement), so a fundamentally new theory would need to be developed. Further, representing uncertainty over the group of nuisance transformations is not easy since these transformations may belong to a discrete set (e.g. permutations) or a Reimannian manifold (e.g. rotations). Developing Bayesian inference algorithms in this setting is nontrivial -- see e.g. https://doi.org/10.1111/j.1467-9868.2010.00765.x
>
> > *The empirical work is somewhat preliminary, in the sense that many of the observations could be followed up in more detail with more substantial studies.*
>
> We agree that the most important contribution of our work is to derive a theoretical framework that enables the neuro/ML community to explore and compare representations in stochastic neural networks. We made a strategic decision to emphasize the broad applicability of this framework, spanning three datasets in biological data, deep generative modeling, and data augmentation / noise robustness. An alternative strategy would have been to do a deep dive into one of these application areas, but we felt that this would not convey the generality of our approach and ultimately lead to a less impactful paper. Given the constraints of a 9-page conference paper we agree that our empirical results are preliminary in each application area, but nevertheless reveal nontrivial insights that are (a) not easily measured with existing methods, and (b) could be the starting point of follow-up studies. We thank the reviewer for acknowledging point (b).
>
> > *It is also limited in the sense that only covariance (pairwise) structure is considered when comparing 'noise' distributions empirically, while the framework would allow for higher-order properties of the distributions to be compared.*
>
> The energy distance ground metric we propose is sensitive to higher-order moments. We repeated several analyses using this approach (Supp. Fig. 4, Supp. Fig 7) and largely found similar (but not identical) results. We agree that a more thorough investigation into the differences between energy distance and Gaussian-Wasserstein distance is warranted in follow-up studies.
>
> There are two main reasons we chose to highlight the 2-Wasserstein ground metric in the majority of our results.
> * First, we think the ability to separately isolate contributions of the mean geometry vs covariance geometry is helpful in several ways. For instance, it enables us to compare our results with existing “deterministic” metrics on trial-average data (i.e. when alpha = 2). The energy distance ground metric includes contributions from higher-order moments, but these are all mixed together into a single distance metric. We think it will be possible for us to disentangle the contributions of these higher moments in future work, but we do not have anything solid to present yet.
> * Second, it is difficult to estimate higher-order moments in biological data and computationally expensive to do so in artificial networks. As a result, there is a large literature in neuroscience on 2nd-order noise correlations, and a smaller (though very interesting!) literature on higher-order correlations (e.g. Ohiorhenuan et al. 2010; https://doi.org/10.1038/nature09178).
>
> Ultimately, we feel like we do not have the space to fully unpack how higher-order moments impact stochastic representational similarity within this paper, but our work provides the theoretical foundation for us to begin tackling this topic.

---

> > ### Comment · Reviewer_yN72 · 2022-12-13
> > **Response**
> >
> > Thanks to the authors for their responses.  I look forward to seeing the follow up work in the directions mentioned.

---

### Comment · Area_Chair_G9k6 · 2022-11-07
**Related Work / Experiments**

I have a few additional points, mainly about experiments, which I find really hard to digest (and I did not get far in it).

The problem of comparing two distributions is very common in machine learning. I think it is worth mentioning the Frechet Inception distance, widely used to assess the quality of generative models (GANs, VAEs, etc). It is the Frechet distance (= W2 distance) between Multivariate Gaussians fitted to the distribution of feature responses of a particular (well trained) neural network w.r.t. inputs drawn from two generators (or datasets) to compare. The differences would be that (5) has an extra expectation over inputs, making it an average of a finer-grained distance, and it has the extra minimization over a group of transformations on top.

For the methods part it is

I find that the experiments are too condensed in the paper and have a significant jump in the concepts and terminology from the main paper. As a result it is very hard to grasp the setup, rationale and the outcomes of each experiment.

### Toy data
Mentioning "representational geometry" appears insufficient, as it does not introduce and explain the setup. It is not obvious that the goal here is to visualize which networks are closer and which ones are further apart, which can be achieved with MDS applied with the proposed metric between feature distributions. Does it need triangle inequality?

The 11 x 9 = 99 networks should result in 11 x 9 pairwise distances matrix. How does that correspond to Fig 2.B? I see 9x9 cells with some 2d gaussian distribution in each.
This experiment does not seem to test any permutations or rotations across the networks. The expectation over the inputs also does not seem to play a role, as for each of the 6 inputs the distribution is just shifted, and so the averaging over $Q$ in (5) has no effect. Why then to include it in the experimental setup, making all the plots more complicated to digest?
I suspect in this case, MDS (or tSNE) on any distance between distributions (normalized by the means) would do the job to discover the 2D layout. The toy experiment therefore makes a rather poor job in illustrating the proposed method.

### Biological Data

The starting question is: "do representations of artificial stimuli (e.g. oriented gratings) provide insight into representations of natural scenes?".
There is a big conceptual jump, and a complicated setup making it hard to understand.
This task does not fit to the method presented this far: the metric was designed to compare representations of different networks for the same (distribution of) stimuli.
Here there are two sets of stimuli presented to the same mouse=network and all repeated for many networks.
The professional slang is also hard to follow. E.g., what is a recording, a recording session, does 31 recordings imply there are 31 mice (= 31 networks)?
What are the metrics computed on? Which group G is used?
Since the experiment probes a small subset of all neurons in each mouse, this again does not map well to the presented method: there may exist no permutation to align the partial representations even if a pair of mice would be identical.

Is the visualization in Fig 3.E useful for anything?

The implication of the comparison seems to be that pairs of mice that had (more) similar representations for gatings did not necessarily had similar representations for images.
But this is possibly compromised by that incompleteness of representations compared.

---

> ### Author Response · Authors · 2022-11-08
> **Quick clarification question on your comment**
>
> Dear AC,
>
> We are preparing a comprehensive revision and response, but wanted to inquire about your concerns regarding "the incompleteness of representations compared". By this are you referring to the issue of only recording a "small subset of all neurons" in each mouse, or something different?
>
> Thank you again for your comments and feedback.

---

> > ### Comment · Area_Chair_G9k6 · 2022-11-09
> > **incompleteness of representations**
> >
> > Dear authors,
> >
> > Yes, that was what I meant. I have a very limited understanding of the experimental setup. I hope that my questions will help to make the paper more accessible and clear regarding possible discrepancies between theory and practice.
> >
> > So what I imagine is that the sensors are placed in the visual cortex in similar positions in all mice, however due to brain differences and imprecision of placement they end up recording a subset of neurons in each case which might have different functions (like some different hypercolumn, sensitive to a different orientation, etc.). So the observed features are 1) incomplete and 2) likely not matchable by any permutation. Then I am wondering whether the search of alignment over permutations is appropriate: if the sensors are in a fixed geometrical layout (like a grid of sensors) or inserted to already maximally match locations, disregarding their order would lose the information.

---

> > > ### Author Response · Authors · 2022-11-09
> > > **Understood**
> > >
> > > Excellent, your understanding of the setup is exactly correct and these are good questions. We should have discussed these limitations in more depth and included a citation to [Shi et al. 2019](https://proceedings.neurips.cc/paper/2019/hash/748d6b6ed8e13f857ceaa6cfbdca14b8-Abstract.html) who provide a detailed investigation on this issue. As you say, this will make the results more accessible to a broader audience, not just neuroscientists who may have encountered this sub-sampling issue before. Stay tuned for our full rebuttal and revision.

---

> ### Author Response · Authors · 2022-11-17
> **Revision Summary**
>
> We are sincerely grateful for the feedback provided by the AC and reviewers. We have worked hard to incorporate all of your suggestions, respond to comments, and clarify areas of minor confusion. The presentation and discussion of our results has been substantially improved thanks to your input. The content of our paper remains effectively the same, with the exception of removing a panel from Figure 3 (it became clear that we tried to squeeze too many ancillary results into a short paper, and by deleting this panel we've streamlined the results).
>
> We believe that the core contribution of our work is to provide a rigorous theoretical framework for quantifying representations in stochastic networks. The reviews seem to largely agree with that this is an important problem and that our work makes a meaningful contribution. The review process helped us realize that our experiments were somewhat condensed, and that take-home messages were not clearly enumerated. We believe our revision is substantially improved in this respect. We have emphasized that our experimental analyses are preliminary and are meant to illustrate a breadth of potential applications. We hope that we have set the stage for future work (including work we have planned ourselves) to do a deeper dive into these outstanding questions.
>
> We respond to the AC comments below, and respond to each reviewer individually.

---

> > ### Author Response · Authors · 2022-11-17
> > **Rebuttal (continued...)**
> >
> > > *The problem of comparing two distributions is very common in machine learning. I think it is worth mentioning the Frechet Inception distance, widely used to assess the quality of generative models (GANs, VAEs, etc).*
> >
> > We agree that quantifying distances between probability distributions is a well-studied mathematical topic and these ideas have been applied elsewhere in machine learning. Our original submission made a good faith effort to acknowledge this literature (e.g. Villani 2009; Szekely & Rizzo 2013, Niles-Weed & Rigollet 2022).
> >
> > But it is a good suggestion that we also include a discussion of Frechet Inception (FI) distance. We have done so in the discussion section of the revision. We note that FI distance is very different from our work both in terms of conceptual motivations and practical applications. You already mentioned a few important differences—the expectation over inputs and the extra minimization over a group of transformations. Most fundamentally, the methods measure different things—FI distance measures the similarity between two stimulus sets, while our work measures similarity between two network representations of the same stimulus set. Importantly, in FI distance, the neural response to each image is deterministic, whereas we are interested in networks that respond stochastically to inputs.
> >
> > > *I find that the experiments are too condensed in the paper and have a significant jump in the concepts and terminology from the main paper.*
> >
> > Since no major concerns were identified in the theory portion of our paper, we have spent the revision period thoroughly revising the experimental results for clarity. We thank the AC and reviewers for helping us identify which portions were confusing and unclear. Specific details are enumerated below.
> >
> > > *It is not obvious that the goal here is to visualize which networks are closer and which ones are further apart, which can be achieved with MDS applied with the proposed metric between feature distributions.*
> >
> > Our revised Figure 2 explicitly labels the four corners of the neural shape manifold and matches them to the four corners of panel A showing neural representations / feature distributions. We think this makes it much more clear.
> >
> > > *Does it need triangle inequality?*
> >
> > One does not necessarily need to satisfy the triangle inequality to run a 2D embedding algorithm like MDS or tSNE. But if triangle inequality violations exist, it is much less clear how to interpret the output. Because we satisfy the triangle inequality we know that there is a true underlying metric space, which may be approximated by a low-D Euclidean embedding.
> >
> > We want to emphasize that satisfying the triangle inequality is not the main point of this figure -- the method we compare to in panel D (see Williams et al., 2021) also satisfies the triangle inequality. The point is that the structure of the underlying data (network representations) is not captured by existing, deterministic methods for quantifying representational similarity methods. The new notions of stochastic representational similarity do recover the intended structure.
> >
> > > *The 11 x 9 = 99 networks should result in 11 x 9 pairwise distances matrix. How does that correspond to Fig 2.B? I see 9x9 cells with some 2d gaussian distribution in each.*
> >
> > We have clarified the text to make it clear that we are working with a 99 x 99 symmetric matrix holding all the pairwise distances. The important part of this figure is the MDS embedding plots, so we have removed the panel showing the raw distance matrix. The block structure in the original figure panel is due to the 2D ground truth structure in the toy models -- each 11x11 sub-matrix holds distances between networks at a fixed covariance scale (diagonal blocks) or at two different scales (off-diagonal blocks).
> >
> > > *This experiment does not seem to test any permutations or rotations across the networks.*
> >
> > This is helpful feedback. Our original intention was to emphasize a different point in this figure -- that the interaction between mean and covariance geometry is captured by our method and not deterministic metrics. However, we can see why it would be confusing to show a dataset where the response distributions (more precisely, the means) are pre-aligned. Thus, we’ve reimagined the figure and shown the toy data both ways -- i.e. in the original configuration and after applying a random rotation to each network. All of our results remain exactly the same of course, since the stochastic shape metrics are invariant to these random rotations. We think this new figure helps the reader build better intuition.
> >
> > **[response continued in next comment]**

---

> > > ### Author Response · Authors · 2022-11-17
> > > **Rebuttal (continued...)**
> > >
> > > > *The expectation over the inputs also does not seem to play a role, as for each of the 6 inputs the distribution is just shifted, and so the averaging over Q in (5) has no effect. Why then to include it in the experimental setup?*
> > >
> > > Having multiple inputs is important to demonstrate that the rotational alignment (over $\mathbf{T} \in \mathcal{G}$) impacts both the geometry of the mean responses and the covariance geometry. If there were just one input condition centered at the origin for each network, then there is no mean response geometry at all and only the covariance ellipse needs to be rotated into alignment. In other words, without more than one input condition, there would be no way to distinguish “positive” from “negative” correlations after rotation is taken into account. Thus, having multiple (at least two) conditions is necessary. We do not believe that modestly increasing the number of image conditions to 6 substantially clutters the figure, and furthermore the color scheme matches Figure 1 (which we have now parenthetically cited in our revision). We prefer to keep this aspect of our experiment as is, since it cues the reader to understand that there are numerous images / conditions to consider.
> > >
> > > > *I suspect in this case, MDS (or tSNE) on any distance between distributions (normalized by the means) would do the job to discover the 2D layout.*
> > >
> > > As described above, we’ve improved the experiment to additionally demonstrate the rotation invariance of the stochastic shape metric. In the new figure it should hopefully be clear that naive distances (e.g. Euclidean distance on the means) would not recover the desired structure.
> > >
> > > > *The starting question is: "do representations of artificial stimuli (e.g. oriented gratings) provide insight into representations of natural scenes?". There is a big conceptual jump, and a complicated setup making it hard to understand.*
> > >
> > > This is good feedback. We have entirely revised this section to make the message simpler and to streamline the introductory paragraph as you’ve suggested. We now list three key take-home messages:
> > > * Across-animal variability in covariance geometry is comparable in magnitude to variability in trial-average geometry
> > > * Across-animal distances in covariance and trial-average geometry are not redundant statistics as they are only weakly correlated
> > > * The relative contributions of mean and covariance geometry to inter-animal shape distances are stimulus-dependent.
> > > Together, we think these findings suggest that the geometry of neural covariance is a potentially interesting and under-explored topic in sensory neuroscience. It could have been the case that the new metrics we developed told the same story as the old metrics (i.e. those only capturing differences in mean geometry). Instead, we find that covariance structure varies across stimuli, is weakly related to the mean response, and responds in interesting ways to experimental manipulations (i.e. changing stimulus statistics).
> > >
> > > We still think that the ratio of covariance-insensitive and mean-insensitive metrics across animals is a useful summary statistic and that it is interesting to note that this ratio changes based on stimulus statistics. However, we place less emphasis on this message (e.g. deleting panel E) in the revision.
> > >
> > > > *Does 31 recordings imply there are 31 mice (= 31 networks)?*
> > >
> > > Yes. Correspondingly, we changed the terminology from “recording session” to “animals” to minimize confusion. We’ve also explicitly defined “neural response” as the number of evoked action potentials. We think that our terminology should be understandable by anyone with some training in neuroscience, and those without this background can skip this section of the paper if needed.
> > >
> > > > *What are the metrics computed on? Which group G is used?*
> > >
> > > We have specified that we used the orthogonal group ($\mathcal{G} = \mathcal{O}$) throughout all of our experiments.

---

> > > > ### Author Response · Authors · 2022-11-17
> > > > **Rebuttal (continued...)**
> > > >
> > > > > *Since the experiment probes a small subset of all neurons in each mouse... there may exist no permutation to align the partial representations even if a pair of mice would be identical.*
> > > >
> > > > This is a very well-known caveat within the neuroscience literature. We apologize we did not cover this in our original submission, but we now discuss it at length in the discussion section. We cite several papers which empirically and theoretically show that these kinds of analyses are meaningful when neurons are subsampled ([Shi et al. 2019](https://proceedings.neurips.cc/paper/2019/hash/748d6b6ed8e13f857ceaa6cfbdca14b8-Abstract.html); [Kriegeskorte & Diedrichsen, 2016](https://doi.org/10.1098/rstb.2016.0278); [Trautmann et al., 2019](https://doi.org/10.1016/j.neuron.2019.05.003)). At a high-level, the theory invokes the JL Lemma to argue that representational geometry is not distorted too badly in practical applications by taking a random projection (i.e. subsampling neurons). The key reason it works is because neurobiological circuits have many redundant components, so one can get a “good enough” sense of the neural representation with a sub-sample. We also provide a new supplemental figure (Supp. Fig. 6) which shows that subsampling neurons quantitatively changes across-animal distances but does not qualitatively change the outcome (the stimulus dependence of the ratio of alpha = 0 to alpha = 2 distances is preserved).
> > > >
> > > > > *Is the visualization in Fig 3.E useful for anything?*
> > > >
> > > > We have deleted Fig 3E to streamline and simplify this section of the results. Thank you for this suggestion.
> > > >
> > > > > *The implication of the comparison seems to be that pairs of mice that had (more) similar representations for gatings did not necessarily had similar representations for images. But this is possibly compromised by that incompleteness of representations compared.*
> > > >
> > > > You have the correct interpretation of our result, and we have confirmed that this conclusion is not compromised by subsampling neurons. However, given your suggestion above (that this analysis is a “conceptual jump”) we have removed the panel in question. We believe this streamlines the narrative substantially, while still leaving plenty of results that demonstrate the importance of our method (see the three “take-home messages” in bullet points above). We apologize for trying to squeeze too much into the original submission.

---

### Decision · Program_Chairs · 2023-01-20

**Decision:**

Accept: poster

**Justification For Why Not Higher Score:**

* The technical contribution is solid -- for a higher score I would reserve more exciting contributions
* experiments demonstrate how the method can be applied, but do not reveal important insights about brain or network function

At the same time I might be overlooking the potential impact, so I give it an accept (poster) with the "decision can be bumped up" flag.

**Justification For Why Not Lower Score:**

High average score; Solid theoretical and methodological contribution; No issues

**Metareview: Summary, Strengths And Weaknesses:**

### Description

One big motivation behind the work is to understand whether the correlations in the noisy signals in the brain are important and need to be accounted for in the comparison of stochastic representations.

The technical contribution is in designing (and efficiently computing) a metric between two stochastic representations that can capture differences in the correlation structure, while at the same time being invariant to a group of transformation. The invariance is needed when the representations in one model are e.g. shifted rotated or permuted wrt to the other. The proposed metric is positioned as a tool for representation geometry. Several such applications to visualize representations were considered: biological data, VAEs, patch-Gaussian data augmentation.

### Evaluation

Reviewers remark that
>the problem of measuring distances between representations in stochastic networks is an important one, both in neuroscience and machine learning,

>the numerical experiments are extensive, the theoretical framework is elegant and will be useful in some applications.

The paper is of high quality, clear and sound. The introduced distance satisfies several desired properties: triangle inequality (useful for interpretability and e.g. nearest neighbour search), consistency with the existing deterministic comparison metrics  Williams et al. (2021) in the deterministic limit.

The distance allows to better quantify similarity of stochastic latent representations. It is proposed to apply it as a tool for visualization and analysis. The paper conducts extensive experiments demonstrating examples of such analysis.
However, there was a concern by several reviewers (and is shared by AC), that it is rather difficult to interpret the results of such analysis and draw some useful takeaways in each case. Although the paper does great job emphasizing interesting observations, the usefulness and implications of these observations are not obvious. E.g. for VAEs a massive experiment (many models) are studied and structured w.r.t. their similarity, however it is not clear what particularly new or important insights it could bring or how it could aid development of better (disentangled) latent representations. E.g. the reviewer yN72 remarks:

>The empirical work is somewhat preliminary, in the sense that many of the observations could be followed up in more detail with more substantial studies. It is also limited in the sense that only covariance (pairwise) structure is considered when comparing 'noise' distributions empirically, while the framework would allow for higher-order properties of the distributions to be compared.

E.g. the reviewer Moyn comments:

>The experiments are useful demonstrations of the method, but don’t seem to shed new light on brain or network function.

And the authors reply:
>It is not the main point of our paper to provide deep insight into new brain/network functions. The main point of our paper is theoretical and methodological. We do feel we've described an important set of analysis methods that have the capability to reveal important future insights into brain/network functions.

It appears to the AC that there is a consensus on the theoretical and methodological value of the work, but the impact as for revealing important insights is not fully demonstrated.


**Note From Pc:**

if the above contains the word "oral" or "spotlight" please see: "oral" presentation means -> notable-top-5% and "spotlight" means -> notable-top-25%. As stated in our emails, we are disassociating presentation type from AC recommendations

**Summary Of Ac-Reviewer Meeting:**

N/A